# TAPIP3D: Tracking Any Point in Persistent 3D Geometry

**Bowei Zhang**[1,2*]    **Lei Ke**[1*]    **Adam W. Harley**[3]    **Katerina Fragkiadaki**[1]
[1]Carnegie Mellon University    [2]Peking University    [3]Stanford University

## Abstract

We introduce TAPIP3D, a novel approach for long-term 3D point tracking in monocular RGB and RGB-D videos. TAPIP3D represents videos as camera-stabilized spatio-temporal feature clouds, leveraging depth and camera motion information to lift 2D video features into a 3D world space where camera movement is effectively canceled out. Within this stabilized 3D representation, TAPIP3D iteratively refines multi-frame motion estimates, enabling robust point tracking over long time horizons. To handle the irregular structure of 3D point distributions, we propose a 3D Neighborhood-to-Neighborhood (N2N) attention mechanism—a 3D-aware contextualization strategy that builds informative, spatially coherent feature neighborhoods to support precise trajectory estimation. Our 3D-centric formulation significantly improves performance over existing 3D point tracking methods and even surpasses state-of-the-art 2D pixel trackers in accuracy when reliable depth is available. The model supports inference in both camera-centric (unstabilized) and world-centric (stabilized) coordinates, with experiments showing that compensating for camera motion leads to substantial gains in tracking robustness. By replacing the conventional 2D square correlation windows used in prior 2D and 3D trackers with a spatially grounded 3D attention mechanism, TAPIP3D achieves strong and consistent results across multiple 3D point tracking benchmarks. Project Page: tapip3d.github.io

## 1 Introduction

Tracking points over time in video—especially through occlusions—has become a valuable tool in robotics and action recognition [20, 19, 12]. Fine-grained motion estimation at the particle level offers a unified framework for capturing temporal changes in object pose, part articulation, deformable structures, and even granular materials. However, most existing point trackers operate either directly in pixel space [20, 19, 12] or in pixel space augmented with depth information [48, 30]. Yet, the predominant source of apparent motion in video is often camera movement, not object motion. Since real-world dynamics unfold in 3D, tracking points directly in 3D space—rather than in the image plane—may be more natural and effective.

In this paper, we ask: Can recent advances in camera pose and depth estimation enable more effective 3D point tracking by explicitly representing and tracking points in 3D, while compensating for camera motion?

To this end, we propose TAPIP3D for **T**racking **A**ny **P**oint **i**n **P**ersistent **3D** Geometry, a 3D point tracking method that represents and iteratively updates multi-frame 3D point trajectories through spatio-temporal attentions in an RGB-D video. Our method represents a video as a spatio-temporal 3D feature cloud. Each point in the cloud represents a 2D feature vector lifted to a corresponding 3D coordinate $(X, Y, Z)$ using sensed or estimated depth. Under known camera motion, we construct

---

*Equal contribution.

39th Conference on Neural Information Processing Systems (NeurIPS 2025).

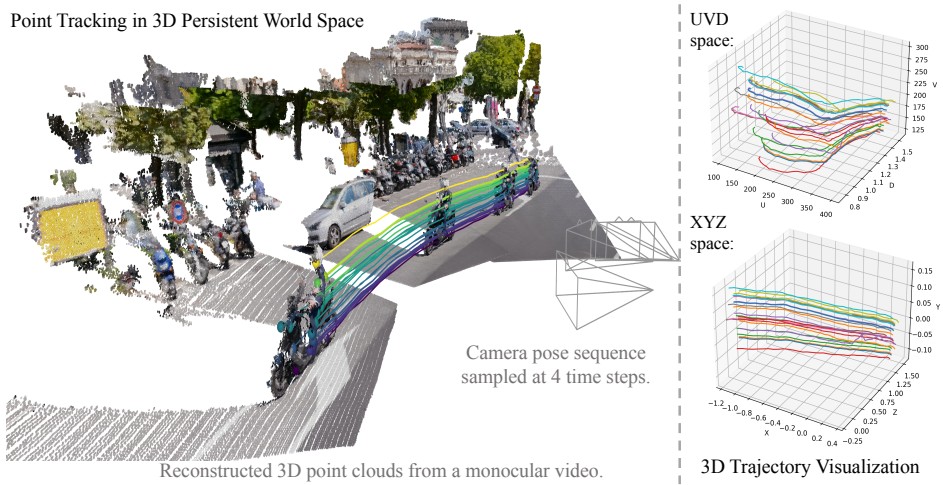

Point Tracking in 3D Persistent World Space

UVD space:

XYZ space:

Camera pose sequence sampled at 4 time steps.

Reconstructed 3D point clouds from a monocular video.

3D Trajectory Visualization

Figure 1: TAPIP3D performs long-term 3D point tracking in a persistent, camera-stabilized world space of 3D feature clouds, surpassing prior methods [48, 30] that operate in camera-dependent UVD (UV pixels + Depth) spaces. Using depth maps and camera poses provided or estimated via MegaSaM [25], TAPIP3D constructs a 3D world space in which camera motion is effectively canceled out. As a result, the 3D trajectories of sampled dynamic points—represented in XYZ world coordinates—are significantly smoother and more linear than those produced in UVD space.

camera-stabilized spatio-temporal feature clouds by applying extrinsics, as shown in Figure 1. Our method performs tracking in either the camera or world 3D feature space by directly featurizing scene coordinates using 3D attentions. Specifically, it identifies the local 3D neighborhood for each query trajectory coordinate as a query group and replaces traditional 2D CNNs with a novel Neighborhood-to-Neighborhood cross-attention design. This mechanism enables the deep feature representation of a query point group at a given timestep to attend to the features of a key point group, i.e., its neighboring 3D points at future timesteps. Moreover, within these cross attentions, we incorporate the 3D relative offset between the query trajectory point and the neighboring target point into the attention values, alongside appearance features, thereby enhancing spatial context awareness.

We test TAPIP3D in established 3D point tracking benchmarks of TAPVid3D [21], LSFOdyssey [40], Dynamic Replica [18] and DexYCB [4] for tracking points in 2D and 3D, using depth from ground-truth, sensors, or estimators, in camera and world 3D coordinate frames. We show TAPIP3D outperforms all previous methods in 3D point tracking metrics, with an especially large margin when accurate depth is available. While existing methods pursue 3D tracking in camera coordinates [30, 48], our experiments show that it is better to track in a "world" coordinate frame, which can be estimated by recent camera and depth estimation methods such as MegaSaM [25]. In addition to showing the impact of coordinate system choices, we ablate the contributions to performance from 3D-centric featurization and quality of depth estimation.

In summary, we present TAPIP3D for 3D point tracking from RGB-D and RGB videos that uses 3D feature clouds for video to estimate 3D point tracks, which demonstrates state-of-the-art performance when ground-truth / sensor depth is available and competitive performance with estimated depth [25]. It can exploit recent advances in depth and camera pose estimation [25, 47] and deliver both camera and world-centric 3D tracks. To our knowledge, TAPIP3D is a pioneering 3D tracking method capable of tracking in a 3D world-centric space, with camera motion factored out. We will release our code upon acceptance.

## 2 Related Work

**2D and 3D Point Tracking**  Recent advances in point tracking formulate the task as the estimation of multi-frame point trajectories, moving beyond traditional pairwise optical flow estimation. Inspired by earlier work on particle video [35], PIPs [12] introduces an iterative refinement approach with dense cost maps. This work along with the Tracking Any Point (TAP) benchmark [7] have catalyzed a series

of advancements [23, 24] including the use of multi-point context [20, 19], enhanced initialization schemes [8], broader correlation contexts [2, 5], and post-hoc densification techniques [22].

Beyond 2D, recent works such as SpatialTracker [48] have extended point tracking to 3D by incorporating depth information. SpatialTracker employs a Triplane representation [3], where 3D featured point clouds per frame from pixel (UV) and depth (D) coordinates are projected onto onto three orthogonal planes. Triplane featurization is faster than our $k$-NN attention, but this speedup comes at the cost of decreased performance. The works of DELTA [30] and SceneTracker [40] also operate on a UVD coordinate system to separately compute appearance and depth correlation features, directly extending the 2D method of CoTracker [20]. DELTA [30] proposes a coarse-to-fine trajectory estimation that manages to estimate dense point tracks across the whole image plane, instead of on a set of sparse locations. In contrast to previous 3D point trackers, TAPIP3D leverages an explicit spatio-temporal 3D feature cloud video representation, instead of a 2.5D one of the image-space and depth, for feature extraction and tracking. By fully utilizing the underlying geometric structure of the scene, our method achieves better point tracking accuracy and consistency. Moreover, our model can estimate trajectories in a 3D world space, by canceling inferred camera motion from the scene image pixels during lifting, which helps performance further.

**Point Tracking via Reconstruction** 3D point tracks can also be extracted from monocular videos through test-time optimization by fitting dynamic radiance fields, such as dynamic NeRFs [33, 41] or Gaussians [29, 42, 36, 6] to the observed video frames supervised from reprojection RGB, depth and motion error. These approaches require per-video optimization and are thus computationally expensive. In contrast, TAPIP3D is a learning-based feed-forward approach for 3D point tracking.

**Scene Flow Estimation** Scene flow estimation extends optical flow into three dimensions by considering pairs of point clouds and estimating the 3D motion between them [11, 13, 34, 39, 27, 46, 10, 31]. Recent works incorporate rigid motion priors, explicitly [39] or implicitly [49] or leveraging diffusion models [26]. Linking 3D scene flow estimates terminates at pixel occlusions by design, similarly to linking 2D flow estimates. Instead our work focuses on inferring multiframe point trajectories through occlusions.

**Learning-Based 3D Foundation Models** Recent methods in learning based camera motion and depth estimation, such as MoGe [44], DUSt3R [45], MonST3R [50], and MegaSaM [25], focus on predicting dense 3D reconstructions from single images, image pairs, or videos, bringing the vision of video to 4D translation closer. MegaSaM [25] delivers highly accurate camera motion and depth by combining learning based initialization and updates with second order optimization, extending earlier work of DROID-SLAM [38]. None of the methods above addresses the problem of estimating the 3D point motion, which is the focus of our work. TAPIP3D builds upon the progress of recent learning-based 3D camera motion and depth estimation methods to explore their use in 3D point tracking, by lifting videos to a 3D world space which makes the scene points easier to track than in the original 2D image plane.

## 3   Method

**Overview.**   TAPIP3D takes as input an RGB-D video, and outputs 3D trajectories for query points, specified by $XYZ$ coordinates. If given camera poses, TAPIP3D outputs trajectories in the coordinate system of the first camera (i.e., "world" coordinates). TAPIP3D begins by featurizing the RGB-D data into per-timestep feature maps, and initializing 3D tracks for the query points, and then iteratively refining the tracks with reference to 3D-based lookups in the features.

Concretely, we take as input an RGB video of $S$ frames $\mathcal{V} = \{\mathbf{I}^t \in \mathbb{R}^{H \times W \times 3}\}_{s=1}^{S}$, a sequence of depth maps $\mathcal{D} = \{\mathbf{D}^t \in \mathbb{R}^{H \times W}\}_{s=1}^{S}$, camera intrinsics (constant over time), and optionally, a sequence of camera poses. The input depth maps can originate from an off-the-shelf monocular depth estimator [44], depth sensors, or GT depth provided by simulation environments. Camera intrinsics and extrinsics can either be estimated algorithmically by video 3D reconstruction models [25] or be provided by the dataset. We are given a set of $Q$ different query points $\mathcal{Q} = \{(X_q^{t_q}, Y_q^{t_q}, Z_q^{t_q}), q = 1 \ldots Q\}$, where $t_q$ represents the timestep at which a query is provided. Queries may also be specified in 2D, in which case we lift them to 3D using depth and camera intrinsics before sending them to the model. For each query, our model produces a 3D point trajectory $\tau_q = (X_q^s, Y_q^s, Z_q^s), s = 1 \ldots S$, and a visibility trajectory $o_q$. The architecture of TAPIP3D is illustrated in Figure 2.

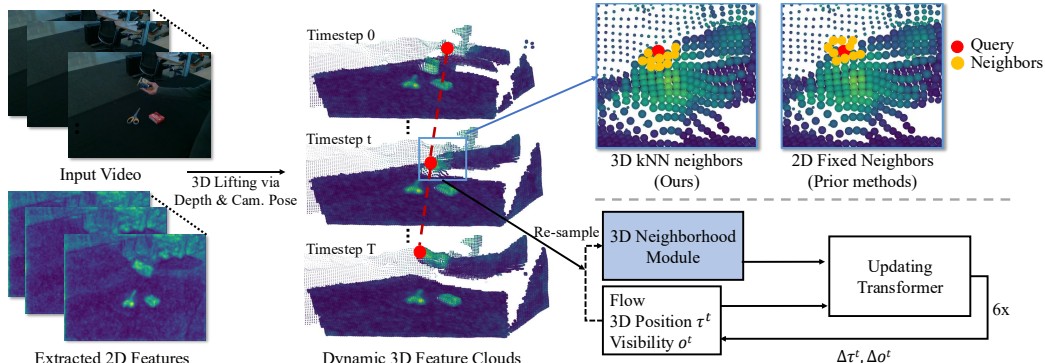

Figure 2: **Architecture of TAPIP3D.** The model takes RGB frames and corresponding 3D point maps as input, computes features from the RGB frames, and transfers them to the 3D points, forming a feature cloud for each timestep. Using camera poses, these feature clouds can be arranged in either world space or camera space. We then apply our 3D neighborhood module (Figure 3) to extract features local to the estimated tracks, followed by a transformer, which iteratively updates the estimated trajectories. *Top right:* Illustration of the difference between 3D $k$-NN (used in our approach) and fixed 2D neighborhoods (used in prior works [19, 5]).

**Tracking in world space vs. camera space.** Previous 3D point trackers [30, 40, 48] represent 3D point trajectories by concatenating depth values to pixel coordinates, which we refer to as $UVD$ space. In contrast, we use $XYZ$ coordinates, which allows us to better leverage 3D priors, and also compensate for camera motion by transforming all frames' points into the first camera's coordinate system (if poses are available). Disentangling camera motion from scene motion is helpful because it makes the 3D tracks more predictable.

## 3.1 Preliminaries: Point tracking as iterative trajectory refinement

TAPIP3D builds on the trend of iterative architectures for point tracking [12, 8, 20, 19]. We build from the most recent such tracker, CoTracker3 [19].

**CoTracker3 overview.** CoTracker3 tracks 2D points across time by iteratively refining their coordinates and visibilities through a transformer. The transformer model is designed to operate on fixed-length sequences of $T$ frames only, and longer videos are processed in a sliding-window manner. Given such a sequence and a set of query points, for each query $q$ it maintains and updates per-frame (2D) locations $\tau_q^t = (u_q^t, v_q^t)$, and confidences $c_q^t$, and visibilities $o_q^t$. The trajectory $\tau_q$ is initialized by copying the query coordinate to all frames (zero motion assumption); $c_q$ and $o_q$ are initialized with zeros. The model then performs $M$ iterations of refinement, producing additive updates to the coordinates and visibilities and confidences.

At each iteration $m$, CoTracker3 samples a patch of 2D features $F_q^t \in \mathbb{R}^C$ around the current location $\tau_q^{t,m}$, and samples a similar 2D patch around the query's reference location, and computes a 4D correlation $\mathrm{Corr}_q^t$ between these two feature patches. This correlation provides signal for how well the image content matches the query, at the current estimated location. This correlation information, along with confidence and visibility estimates from the previous step, and position embeddings for motion and time, form a token for each timestep of each track:

$$G_q^t = [\mathrm{Corr}_q^t, c_q^t, o_q^t, \gamma(\tau_q^t - \tau_q^{t-1}), \gamma(\tau_q^{t+1} - \tau_q^t), \gamma(t)], \tag{1}$$

where $\gamma$ is a Fourier-based position embedding function. The trajectory tokens are then updated through spatio-temporal attentions, which predict updated point positions, confidences and visibilities. At test time, to track points in a video longer than $T$ frames, CoTracker3 advances the inference window by $\frac{T}{2}$ frames, and initializes the first half of the trajectories with the estimates of the previous window, and then repeats inference.

## 3.2 Tracking in a Persistent 3D Geometric Space

**Video representation as 3D-enriched 2D maps.** Our key innovation is to represent the input video as a sequence of multi-scale 2D feature maps enriched with *3D coordinates* [14, 15]: each 2D cell holds both a $C$-dimensional feature vector and a 3D $(X, Y, Z)$ position, where the 3D positions come from unprojected depths. This representation allows us to treat the data as a sequence of featurized point clouds (or "feature clouds"), despite their 2D grid arrangement in memory. Compared to other 3D scene representations such as tri-plane encoders [48] and UVD coordinates [30], our representation encodes 3D geometry more explicitly and without distortions or approximations. We first encode each video frame with a 2D image encoder with downsampling rate $\ell$, producing a sequence of feature maps at a reduced resolution $\frac{H}{\ell} \times \frac{W}{\ell}$, where $W$ and $H$ are the original width and height of the image. We convert these into $L$ levels of multi-scale features via 2D downsampling: $\mathcal{F}_l = \{\mathbf{F}^{l,t} \in \mathbb{R}^{\frac{H}{\ell 2^{l-1}} \times \frac{W}{\ell 2^{l-1}} \times (3+C)}\}_{t=1}^T$, for $l = 1, \ldots, L$. Our downsampling uses nearest-neighbor interpolation for coordinates and average pooling for features.

**3D Neighborhood-to-Neighborhood Attention.** Our next step is to contextualize each timestep of each trajectory ($\tau_q^t$) with information local to it in the input data. Previous methods compute correlation features for each trajectory point using 2D neighborhoods, either on the image plane [20, 5] or on triplanes [48]. Our representation enables us to use 3D neighborhoods at this step, which lets the model take advantage of the fact that the targets are in a 3D world and are often disambiguated by 3D distance. Our overall strategy is illustrated in Figure 3.

We remind the reader that there are $Q$ queries, and the model operates on a window of $T$ timesteps at once.

We begin by supplementing each query point with features from a lo-

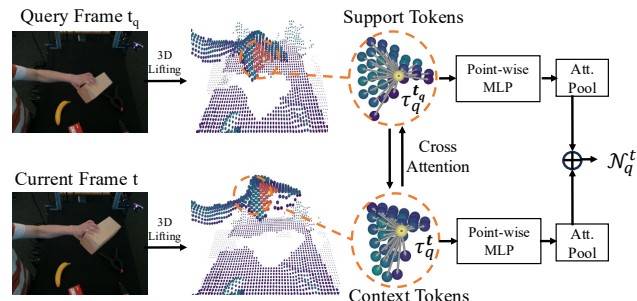

Figure 3: **3D Neighborhood-to-Neighborhood (N2N) attention.** Given a 3D query point at a specific timestep, we first identify its local 3D neighbors using $k$-NN to form a support group. Then, within the point cloud at another timestep, we find another $K$ nearest 3D neighbors to construct *context tokens*. We apply bi-directional cross-attention between the support tokens and context tokens to capture spatio-temporal correspondences.

cal neighborhood of points. For a trajectory $\tau_q$ whose query originates in frame $t_q$, we retrieve $K$ nearest neighbors from the feature cloud of that timestep (e.g., $K = 32$), using 3D distances to find neighbors. For each neighbor, we also store its relative offset from the query, and compute a positional embedding from this offset. These $K$ features, with their position embeddings, serve as "support" tokens for the query, capturing shape information about the query. In total, we have $Q \cdot K$ support tokens.

Next, for each trajectory, at each timestep, we use the current estimated coordinate ($\tau_q^t$) to obtain $K$ nearest neighbors from that timestep's feature cloud. For each neighbor, we also store its relative offset from the estimated coordinate ($\tau_q^t$), and compute a positional embedding from this offset. These $K$ features, with their position embeddings, serve as "context" tokens, capturing 3D-localized information about the scene at that timestep. In total, we have $Q \cdot T \cdot K$ context tokens.

We now have a bag of "support" tokens available for each track, and a bag of "context" tokens available for each track at each timestep, and we would like to compute information about how well each track is corresponding at every timestep. For each query, we copy its "support" bag to all timesteps (making a total of $Q \cdot T \cdot K$ support tokens), and then perform bi-directional cross-attention between each "support" bag and its corresponding "context" bag, followed by a per-token MLP. We then compress each bag into a single token, via attention pooling: free-variable tokens (one per bag type) cross-attend to the bags, retrieving their contents, yielding a total of $Q \cdot T \cdot 2$ tokens (two summaries per query per timestep). We reduce this to $Q \cdot T$ summaries with a sum.

We perform this 3D neighborhood-to-neighborhood contextualization for all scale levels in parallel, yielding a total of $Q \cdot T \cdot L$ tokens. We concatenate across scales, to produce $Q \times T$ vectors capturing the multi-scale neighborhood information, which we denote as $\mathcal{N}_q^t$.

**3D trajectory updating transformer**    To complement the multi-scale neighborhood features with information about the trajectory itself, we concatenate sinusoidal embeddings of motion vectors implied by the trajectory, sinusoidal embeddings of 2D coordinates implied by the trajectory, occlusion estimates, and a sinusoidal embedding of the timestep, creating general-purpose trajectory tokens $G_q^t$:

$$G_q^t = [\mathcal{N}_q^t, \gamma(\tau_q^t - \tau_q^{t-1}), \gamma(\tau_q^{t+1} - \tau_q^t), \gamma(\pi_t(\tau_q^t)), o_q^t, \gamma(t)], \tag{2}$$

where $\gamma$ denotes the Fourier encoding function and $\pi_t : \mathbb{R}^3 \to \mathbb{R}^2$ denotes camera projection. Pixel coordinates are explicitly included to assist the model in identifying points located outside the image boundaries.

These trajectory tokens are processed by a transformer utilizing proxy tokens, following Co-Tracker3 [19]. That is, we use virtual trajectories (that do no correspond to any query points) to help approximate spatial and temporal attention across our main $Q \cdot T$ tokens. The transformer outputs additive updates for positions and visibility scores:

$$\tau_q^t = \tau_q^t + \Delta\tau_q^t, \quad o_q^t = o_q^t + \Delta o_q^t \tag{3}$$

which serve as inputs for subsequent iterations.

**Training**    We train with four iterative inference steps, and supervise each iteration's output. Let $\{\tau_q\}_{q=1}^Q$ denote the predicted trajectories, and let $\{\tilde{\tau}_q\}_{q=1}^Q$ denote the ground truth trajectories, and similarly define $\{o_q\}_{q=1}^Q$ and $\{\tilde{o}_q\}_{q=1}^Q$ for visibility. We define the loss for one iteration as:

$$\mathcal{L} = \sum_{q=1}^Q \sum_{t=1}^T \frac{1}{d_i^t} \left\| \tau_q^t - \tilde{\tau}_q^t \right\|_2 + \alpha_{\text{vis}} \text{CE}(o_q^t, \tilde{o}_q^t) \tag{4}$$

where $d_q^t$ denotes the depth of $\tau_q^t$, and $\frac{1}{d_q^t}$ scales down the loss of far-away points. CE denotes the binary cross entropy loss and $\alpha_{\text{vis}}$ is a weighting factor balancing the visibility loss with the tracking loss. During training, we augment the training data by applying random rigid transforms to each frame's point cloud.

**Implementation Details**    Our model is trained on the Kubric MOVi-F dataset [9]. We initialize the image encoder with CoTracker3's pre-trained weights [19]. We train on 8 L40S GPUs with a batch size of 1 for 200K iterations, which takes roughly 4.2 days and consumes only 20GB VRAM with BF16 mixed precision. We train and test with a window length of $T = 16$. We train with videos of length 24 (making two windows per batch), and 384 trajectories per sample. We scale the coordinates in the point maps to unit variance before processing each window to simplify learning. We optimize using AdamW [28] with the learning rate and weight decay both set to 5e-4. We use a learning rate schedule with 10k warmup steps followed by cosine annealing. During inference, under BF16 mixed precision, our model achieves a speed of 10 FPS and consumes around 2.6GB of VRAM when tracking 1024 query points across 32 frames on an L40S GPU.

## 4   Experiments

We evaluate TAPIP3D on both 3D and 2D point tracking benchmarks, on videos from simulated and real-world scenes with various depth modalities, including sensor depth, estimated depth, and simulator depth. Our 2D estimates are obtained by projecting the inferred 3D point tracks onto the image plane. Our experiments aim to answer the following questions: **(1)** How does TAPIP3D compare with the state of the art in 3D and 2D tracking? **(2)** How does depth quality (estimated versus GT) affect performance of TAPIP3D? **(3)** How does camera vs. world space affect performance? **(4)** How does 3D neighborhood-to-neighborhood attention compare against other contextualization strategies?

**Evaluation metrics**    We adopt the 3D point tracking metrics from **TAPVid-3D** [21]: **APD**$_{3D}$ ($< \delta_{\text{avg}}$) quantifies the average percentage of visible points whose 3D positional errors fall within

depth-adaptive thresholds that scale with the ground-truth depth. **Occlusion Accuracy (OA)** measures the correctness of visibility predictions, while **Average Jaccard (AJ)** jointly evaluates position and visibility accuracy through the intersection-over-union between correctly predicted visible points and ground truth.

Table 1: Comparison of long-term 3D point tracking methods on the large-scale real-world TAPVid-3D [21] benchmark. We take the estimated depth from MegaSAM [25]. For **TAPIP3D-world**, we also leverage the camera pose estimation from MegaSAM [25]. Note that ADT [32] has significant camera motions while PStudio [17] has a static camera. M-SaM: depth and camera poses estimated by MegaSaM [25].

| Methods | ADT | | | DriveTrack | | | PStudio | | | Average | | |
|---|---|---|---|---|---|---|---|---|---|---|---|---|
| | $AJ_{3D} \uparrow$ | $APD_{3D} \uparrow$ | OA↑ | $AJ_{3D} \uparrow$ | $APD_{3D} \uparrow$ | OA↑ | $AJ_{3D} \uparrow$ | $APD_{3D} \uparrow$ | OA↑ | $AJ_{3D} \uparrow$ | $APD_{3D} \uparrow$ | OA↑ |
| CoTracker3 [19] + M-SaM | 20.4 | 30.1 | 89.8 | 14.1 | 20.3 | **88.5** | 17.4 | 27.2 | 85.0 | 17.3 | 25.9 | 87.8 |
| SpatialTracker [48] + M-SaM | 15.9 | 23.8 | 90.1 | 7.7 | 13.5 | 85.2 | 15.3 | 25.2 | 78.1 | 13.0 | 20.8 | 84.5 |
| DELTA [30] + M-SaM | 21.0 | 29.3 | 89.7 | 14.6 | **22.2** | 88.1 | 17.7 | 27.3 | 81.4 | 17.8 | 26.3 | **86.4** |
| **TAPIP3D-camera** + M-SaM | 21.6 | 31.0 | 90.4 | 14.6 | 21.3 | 82.2 | 18.1 | 27.7 | 85.5 | 18.1 | 26.7 | 86.0 |
| **TAPIP3D-world** + M-SaM | **23.5** | **32.8** | **91.2** | **14.9** | 21.8 | 82.6 | **18.1** | **27.7** | **85.5** | **18.8** | **27.4** | **86.4** |

**Datasets** We evaluate the performance of our model and baselines in the following synthetic and real world datasets: **1) TAPVid3D** [21] integrates videos from three distinct real-world datasets covering diverse scenarios: DriveTrack [1], Panoptic Studio [17], and Aria Digital Twin [32]. Together, these datasets provide a total of 4,569 evaluation videos, with video lengths ranging from 25 to 300 frames. **2) DexYCB-Pt**: a dataset we introduce by leveraging ground-truth object and hand poses from manipulation scenes in DexYCB [4] to generate accurate 3D point tracks on 8,000 real-world RGB-D videos. **3) LSFOdyssey** [40]: a synthetic dataset that contains 90 videos with complex but realistic long-range motion using humanoids, robots, and animals. Each video is 40 frames long and has 3D point trajectories and occlusion labels, sourced from PointOdyssey [51]. **4) DynamicReplica** [18]: a synthetic 3D dataset that contains 500 videos of articulated human and animal models. Each video is 300 frames long.

**Baselines** We compare TAPIP3D against the following 2D and 3D point trackers: **1) DELTA** [30]: a very recent state-of-the-art 3D point tracker that uses depth as an additional input for computing cross-correlations, essentially extending CoTracker [20] to a UVD space. **2) SpatialTracker** [48]: a 3D point tracker that builds upon CoTracker's architecture and uses a tri-plane representation to featurize $(u, v, d)$ coordinates. **3) CoTracker3** [19], which operates in 2D, but we lift its trajectories to 3D using the depth at the trajectory coordinates.

We consider the following versions of our model:

1) *TAPIP3D-world*: This is TAPIP3D tracking in world coordinates, by using the camera pose trajectory to cancel camera motion in the 3D coordinates.

2) *TAPIP3D-camera*: This is TAPIP3D tracking in camera coordinates, where information of camera pose is not used, and thus the observed 3D motion reflects the combination of camera motion and scene motion.

In all the evaluations, our model is trained on the Kubric MOVi-F dataset [9]. TAPIP3D-world and TAPIP3D-camera adopt the same trained checkpoint but conduct inference in the two different 3D coordinate systems.

## 4.1 Point Tracking Evaluation

We first evaluate TAPIP3D against strong baseline methods [19, 48, 30] on the challenging real-world TAPVid-3D [21] benchmark. Table 1 summarizes the comparative 3D tracking performance, utilizing depth and camera poses obtained from state-of-the-art 3D reconstruction models [25] for each frame. TAPIP3D consistently outperforms recent state-of-the-art methods, including SpatialTracker [30] and DELTA [48], across all three subsets in terms of $AJ_{3D}$. We also visualize the 3D point tracking results in Figure 4; 3D point tracks from TAPIP3D are noticeably more accurate than those of the baselines. Notably, on the ADT dataset [32], characterized by particularly challenging camera motions, TAPIP3D-world significantly surpasses DELTA [30], achieving improvements of 2.5 points in $AJ_{3D}$ and 3.5 points in APD3D. Furthermore, TAPIP3D-world demonstrates a clear advantage

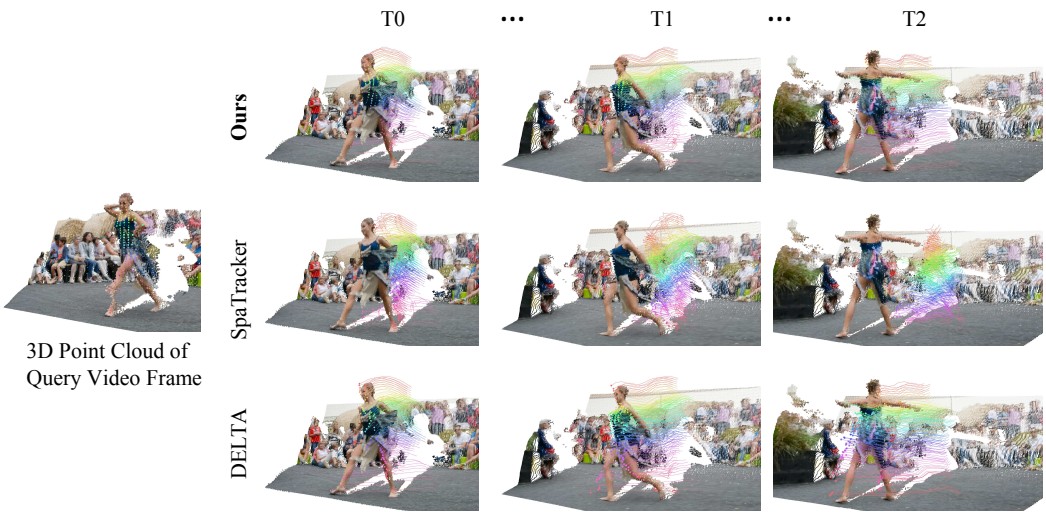

Figure 4: **Visual results comparison.** We qualitatively compare the 3D point tracking results between TAPIP3D with state-of-the-art approaches SpaTracker [48] and DELTA [30] under the same 3D point cloud produced by MegaSaM [25], where we observe serious point drifting issues of SpaTracker and DELTA under large 3D object motion. Zoom in for better view.

over TAPIP3D-camera, highlighting the benefits of conducting tracking in a persistent 3D world space rather than in camera-centric coordinates.

Next, we present results for 3D point tracking on the real-world manipulation dataset DexYCB [4], as shown in Table 3. TAPIP3D notably boosts performance over the previous best method [30], improving $AJ_{3D}$ from 26.4 to 30.3. Additionally, we observe that SpaTracker [48] achieves limited performance, even falling below the depth-lifted CoTracker3 [19]. This suggests that SpaTracker's triplane representation struggles to leverage the accurate sensor depth provided by DexYCB effectively.

Table 2: Comparison of long-term 3D and 2D point tracking results in LSFOdyssey [40] and Dynamic Replica [18] benchmarks with improved input depth quality ( MegaSAM [25] → GT Depth). The best results per column are highlighted in **bold**. GT denotes using GT depth and camera poses provided by the dataset. M-SaM: depth and camera poses estimated by MegaSaM [25].

| Methods | LSFOdyssey | | | | | Dynamic Replica | | | | |
|---|---|---|---|---|---|---|---|---|---|---|
| | $AJ_{3D}\uparrow$ | $APD_{3D}\uparrow$ | $AJ_{2D}\uparrow$ | $APD_{2D}\uparrow$ | $OA\uparrow$ | $AJ_{3D}\uparrow$ | $APD_{3D}\uparrow$ | $AJ_{2D}\uparrow$ | $APD_{2D}\uparrow$ | $OA\uparrow$ |
| CoTracker3 [19] + M-SaM | 19.0 | 28.1 | 75.0 | 88.0 | **88.5** | 20.0 | 30.2 | 61.4 | 80.0 | **86.2** |
| SpatialTracker [48] + M-SaM | 14.8 | 22.8 | 73.8 | 88.5 | 85.7 | 6.6 | 12.0 | 54.1 | 72.8 | 81.7 |
| DELTA [30] + M-SaM | 18.9 | 28.0 | 75.2 | 90.2 | 83.7 | 18.0 | 27.2 | 61.3 | 77.3 | 81.0 |
| **TAPIP3D-camera** + M-SaM | 20.2 | 28.6 | 70.4 | 86.2 | 83.8 | 20.6 | 30.3 | 55.3 | 77.1 | 80.2 |
| **TAPIP3D-world** + M-SaM | 20.5 | 29.2 | 72.3 | 87.5 | 83.6 | 20.6 | 30.2 | 56.1 | 77.7 | 78.6 |
| CoTracker3 [19] + GT | 28.4 | 35.0 | 75.0 | 88.0 | **88.5** | 27.4 | 38.0 | 61.4 | 80.0 | **86.2** |
| SpatialTracker [48] + GT | 8.2 | 13.3 | 70.6 | 85.9 | 84.9 | 7.6 | 13.6 | 56.4 | 74.7 | 83.8 |
| DELTA [30] + GT | 37.7 | 50.1 | 72.4 | 88.4 | 82.3 | 27.4 | 37.7 | 65.6 | 80.5 | 83.8 |
| **TAPIP3D-camera** + GT | 68.3 | 83.2 | 76.0 | 91.2 | 86.2 | 53.7 | 70.8 | 64.6 | 84.7 | 84.7 |
| **TAPIP3D-world** + GT | **72.2** | **85.8** | **78.5** | **92.8** | 86.9 | **55.5** | **72.8** | **66.2** | **85.7** | 85.3 |

Lastly, we evaluate our model's 3D point tracking performance on the synthetic LSFOdyssey [40] and DynamicReplica [18] benchmarks, as shown in Table 2. Comparisons are conducted between our method and baselines under different depth sources: ground-truth depth from simulation, and estimated depth from MegaSaM [25]. We report two notable findings: **1)** *TAPIP3D-world outperforms all competing 3D point trackers* in $AJ_{3D}$ when provided with depth estimation from either MegaSaM [25] or ground-truth depth. Moreover, utilizing ground-truth depth, our approach even outperforms the 2D tracking accuracy of CoTracker3 [19]. **2)** As the depth quality improves (MegaSaM [25] → GT Depth), the performance enhancement of our method in $APD_{3D}$ is substantially greater compared to DELTA [30] and SpaTracker [48], underscoring the effectiveness of our 3D point feature representation in world coordinates.

Table 3: 3D point tracking comparison on DexYCB-Pt using Sensor Depth (SD).

| Methods | DexYCB-Pt | | |
| --- | --- | --- | --- |
| | $AJ_{3D}$ ↑ | $APD_{3D}$ ↑ | OA↑ |
| CoTracker3 + SD [19] | 14.9 | 26.1 | 70.9 |
| SpatialTracker [48] | 5.5 | 11.4 | 66.8 |
| DELTA [30] | 26.4 | 43.3 | **72.8** |
| **TAPIP3D (Ours)** | **30.3** | **52.4** | 71.3 |

Table 4: Ablation on the searching neighbors methods when performing correlation. We compare 3D $k$-NN vs. Fixed 2D Neighbors in DexYCB-Pt benchmark using Sensor Depth (SD).

| Methods | $AJ_{3D}$ ↑ | $APD_{3D}$ ↑ | OA↑ |
| --- | --- | --- | --- |
| Fixed Neighbors in 2D | 27.7 | 50.0 | 67.7 |
| **$k$-NN in 3D** | **29.8** | **51.6** | **70.9** |

Table 5: Ablation experiments of TAPIP3D. To isolate the influence of depth and camera pose, we use the annotations provided by the LSFOdyssey dataset [40]. **N2N Att.**: Neighborhood-to-Neighborhood Attention. **Cam**: 3D camera space. **World**: 3D world space. Note that the DexYCB-Pt video dataset is with static camera such that the result number for **Cam** and **World** is the same.

| Coord. Systems | LSFOdyssey | | |
| --- | --- | --- | --- |
| | $AJ_{3D}$ ↑ | $APD_{3D}$ ↑ | OA↑ |
| UV + D | 63.4 | 77.0 | **87.0** |
| UV + log(D) | 62.9 | 77.9 | 84.1 |
| **XYZ (Cam)** | 67.1 | 81.6 | 85.8 |
| **XYZ (World)** | **70.7** | **84.1** | 86.6 |

(a) 3D Coordinates Systems

| Methods | LSFOdyssey | | | DexYCB-Pt | | |
| --- | --- | --- | --- | --- | --- | --- |
| | $AJ_{3D}$ ↑ | $APD_{3D}$ ↑ | OA↑ | $AJ_{3D}$ ↑ | $APD_{3D}$ ↑ | OA↑ |
| Cam w/o N2N Att. | 59.4 | 72.7 | **88.0** | 26.3 | 46.6 | 68.8 |
| **Cam w N2N Att.** | **67.1** | **81.6** | 85.8 | **29.8** | **51.6** | **70.9** |
| World w/o N2N Att. | 62.1 | 75.1 | **88.2** | 26.3 | 46.6 | 68.8 |
| **World w N2N Att.** | **70.7** | **84.1** | 86.6 | **29.8** | **51.6** | **70.9** |

(b) Ablation on **N2N** Attention

## 4.2 Ablations

In Tables 4 and 5 we ablate our model's design choices regarding the selection of nearest neighbors in 3D vs. 2D space, UVD vs. XYZ space for point tracking and neighborhood-to-neighborhood attention vs. conventional point-to-neighborhood cross attention for feature extraction. We draw following conclusions:

**Computing nearest neighbors in 3D space is better than fixed 2D.** In Table 4, we see that our 3D $k$-NN design improves the $AJ_{3D}$ metric from 27.7 to 29.8. This indicates that our 3D $k$-NN mechanism leverages 3D geometric information to effectively filter out irrelevant 2D neighbors during correlation, as illustrated in Figure 2.

**Tracking in world space helps.** In Table 5a, TAPIP3D that tracks in XYZ (world) and XYZ (Cam) space outperform versions that track in UV+D and UV+log(D) space, the latter two commonly adopted in existing state-of-the-art methods [20, 30]. Specifically, tracking in world XYZ coordinates performs best ($AJ_{3D}$ metric). Tracking in world coordinates generalizes effectively to scenarios with significant camera motion, such as those in LSFOdyssey [40].

**Considering support points during attentions helps**. In Table 5b, we compare the proposed Neighborhood-to-Neighborhood Attention mechanism against conventional point-to-neighborhood cross attention. Specifically, we replace the Neighborhood-to-Neighborhood Attention, which groups each query point with its neighboring context points in both spatial dimensions, with standard 2D attention that considers only the individual query points. Our experiments show that Neighborhood-to-Neighborhood Attention significantly improves the $APD_{3D}$ metric from 75.1 to 84.1. This region-to-region cross-attention approach notably mitigates matching ambiguities by incorporating richer context compared to the simpler point-to-neighborhood matching baseline. Similar conclusion was reached in LocoTrack [5] but for 2D image plane using CNNs, which we extend to 3D feature clouds using cross-attention.

## 5   Limitations

Despite achieving state-of-the-art results, TAPIP3D's performance can be affected by the fidelity of the input depth maps. As a model that works in XYZ space, it expects the input geometry to remain stable and consistent across frames to ensure well-defined 3D trajectories. While MegaSaM generally provides reliable geometry, failures may occur in scenes with extreme depth variations or small, distant elements that appear blurred. In such cases, depth flickering or incorrect surface connections may arise, leading to degraded tracking quality. Furthermore, when high-quality depth

(e.g., from sensors) is not available, TAPIP3D may perform worse on 2D metrics than UVD-space or purely 2D tracking methods, since its 2D trajectories are projected from 3D estimates and rely on geometrically consistent data. This limitation can be mitigated via depth map pre-processing methods such as completion and noise filtering. With ongoing advancements in 3D vision reconstruction models offering increasingly accurate priors, we anticipate continued improvements in the robustness of our method.

# 6    Conclusion

We introduced TAPIP3D, a novel approach for multi-frame 3D point tracking that represents videos as spatio-temporal 3D feature clouds in either a camera-centric or world-centric coordinate frame, and employs neighborhood-to-neighborhood attention to contextualize the estimated tracks in the feature clouds, and iteratively updates the trajectories. By lifting 2D video features into a structured 3D space using depth and camera motion information, TAPIP3D addresses fundamental limitations of previous 2D and 3D tracking methods, particularly in handling large camera motion. We validated TAPIP3D on established synthetic and real world tracking benchmarks demonstrating state-of-the-art performance in 3D point tracking when ground-truth depth is available and competitive performance with estimated depth. By leveraging advances in depth estimation and camera pose prediction, TAPIP3D paves the way for more robust and accurate multi-frame tracking in both RGB and RGB-D videos.

**Acknowledgements**

This material is based on work supported by an NSF Career award, an Amazon faculty award, and AFOSR Grant FA9550-23-1-0257.

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

# Supplementary Material

In this supplementary material, Section A provides further training details of our model, including a memory design that significantly helps reduce VRAM usage. Section B explains the details of our evaluation setup. In Section C, we provide details regarding the inference speed of our model. In Section D, we present additional visualization of our method using stereo depth, enabled by FoundationStereo [47], a powerful stereo matching method. Section E provides further ablation results to offer insights into model behavior. Additionally, we report the evaluation results of our model on TAPVid-DAVIS [7] in Section G. For extensive visual comparisons of our results in Section H, please refer to our project page.

## A    Additional Training Details

Table 6 lists the hyperparameters used during training. We adopt all the data augmentations from PIPs [12] (also used in CoTracker3 [19]), including random Gaussian blur, random occlusion, random cropping, and color jittering, along with a smoothly varying sequence of random rigid transformations applied on the input point clouds to improve camera-centric performance. During training, we perform $M_{\text{train}} = 4$ iterative refinements per window. The loss at the $m$-th iteration is weighted by $\gamma^{M_{\text{train}}-m}$, where $\gamma$ is the discount factor. The total loss for each sample is computed as the sum of these discounted losses across all iterations and windows. To mitigate unstable gradients in the early training stage, we additionally scale the total loss by 0.005.

Table 6: Training hyperparameters

| Hyperparameter | Value |
|---|---|
| Learning rate | 0.0005 |
| Weight decay | 0.0005 |
| Iteration refinements ($M_{\text{train}}$) | 4 |
| LR schedule | OneCycleLR |
| Training steps | 200,000 |
| Batch size | 8 |
| Optimizer | AdamW |
| Max grad norm | 10.0 |
| Visibility loss weight ($\alpha_{\text{vis}}$) | 3.0 |
| Loss discount factor ($\gamma$) | 0.8 |
| Total loss multiplier | 0.005 |

During training, we observed that the neighborhood-to-neighborhood attention module consumes a large amount of GPU memory, as memory usage accumulates over the course of iterative predictions. To reduce this VRAM overhead, we introduced a memory-saving strategy that significantly helps reduce VRAM usage. Specifically, we detach the gradients of the predicted coordinates and visibilities after each iteration, allowing the loss for each window and iteration to be computed and backpropagated independently. We immediately backpropogate the loss of each window as soon as it is ready, and release the part of the computation graph that is no longer needed. The only component that needs to retain a shared computation graph across iterations is the image encoder. To avoid repeated backpropagation through it, we truncate the gradient at the image features and accumulate gradients there. These accumulated gradients are then backpropagated once, after the final iteration of each batch.

This strategy ensures that VRAM usage remains constant regardless of the number of iterations and windows. Despite its simplicity, it has minimal impact on training speed and does not sacrifice model performance, while significantly reducing memory usage—from over 48GB (which previously caused GPU memory overflow) to approximately 20GB.

## B    Evaluation Details

For evaluation, we use the metrics defined in TAPVid [7] and TAPVid-3D [21] throughout the paper. All baselines, as well as our method, operate in a window-by-window manner, so they can only predict trajectories after the query frame (i.e., in the forward direction). However, the TAPVid-3D metric also requires evaluating trajectories before the query frame (i.e., in the backward direction). Therefore, to obtain these backward trajectories, we reverse the video in time and run inference again.

Following previous 2D tracking methods [19, 8], we augment the provided query points with additional support queries sampled from a 2D grid in the first frame. These points are lifted into 3D and inferred jointly with the original queries. The grid resolution is set to $16 \times 16$ for all models. For datasets with known camera intrinsics, we use the ground-truth intrinsics both to lift image features into 3D feature clouds and to convert the outputs of UVD-space models into XYZ coordinates. We do not apply the trick of treating low-confidence point as occluded, as done in CoTracker3 [19].

Our model additionally requires a scale factor to normalize the coordinates when constructing spatio-temporal feature clouds. During training, we use the standard deviation of all 3D points in the feature clouds as the normalization factor. To reduce the influence of extreme values from depth estimates during evaluation, we assume a rough upper bound for the depths of interest, which is computed as twice the maximum depth of ground truth tracks. We then compute the scale factor as the standard deviation of 3D points whose depths fall within the upper bound. We note that points lying out of the depth upper bound are not excluded from the feature clouds and can still be tracked by the model.

## C    Inference Speed

To assess the efficiency of our model, we measure its inference speed on a single L40S GPU using BF16 mixed precision. We evaluate the model on 32-frame video sequences, where 1,024 query points are provided in the first frame. Our model achieves an inference speed of 11.3 FPS. It is approximately $1.3\times$ slower than SpaTracker [48] on the same videos. It is of course substantially faster than optimization-based methods such as Shape-of-Motion [42], which require several hours of inference-time optimization for a 300-frame sequence.

For monocular videos, we find that generating depth maps and estimating camera poses are the most time-consuming components in our pipeline. For instance, processing a sequence from DexYCB takes 121.1s for geometry estimation with MegaSaM, yet our model only requires 6.4s for point tracking. Among this, feature extraction accounts for 0.11s, encoding trajectory tokens for 6.04s, and the updating transformer for 0.25s. We also observe that our model is more robust to degraded camera pose quality than to degraded depth estimation. Consequently, one potential way to accelerate the pipeline is to use a more efficient depth estimator, such as CUT3R [43], especially in scenarios where sensor depth is available. We expect that future improvements in depth estimation models will help reduce this bottleneck.

## D    Inference with Stereo Videos

While our method requires reasonably accurate unprojected point clouds as input, obtaining high-quality depth maps remains challenging in many real-world scenarios. To mitigate the limitation, we demonstrate how our method can be combined with FoundationStereo [47], a recent advancement in stereo matching, to produce metric-scaled point tracks from in-the-wild stereo videos.

Specifically, we follow the pipeline proposed in Stereo4D [16] to download stereo videos in the VR180 format from YouTube and rectify them. We then estimate disparity maps frame by frame using FoundationStereo [47], using them to unproject the image features into camera-centric spatio-temporal feature clouds. Figure 5 shows two examples produced by this process. Comparison videos are provided in the supplementary material (TAPIP3D-supp.mp4). While the examples are

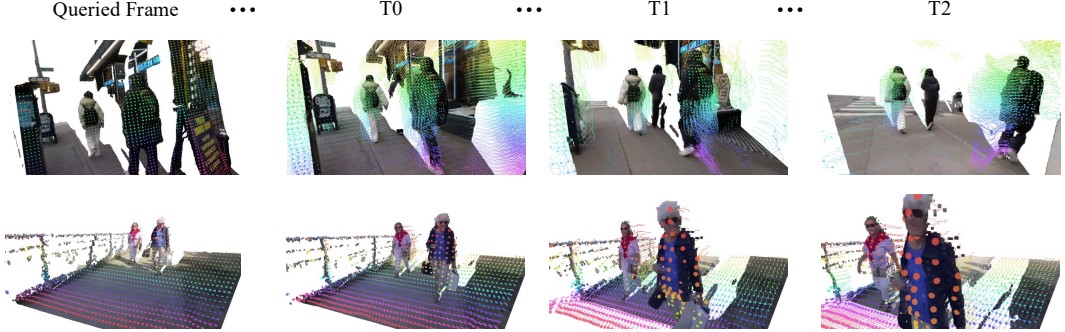

Figure 5: Examples of running our model with depth maps estimated by FoundationStereo [47]. Tracking is performed in the camera-centric frame, so the trajectories above include both camera and object motion. Refer to our videos on the project page for better visual comparison.

camera-centric tracking results, one can also combine the estimated disparity with camera poses from methods such as MegaSaM [25] to track in world-centric coordinates.

## E  Additional Ablations and Analyses

### E.1  Ablation on Neighborhood Size

We train our model with $K = 32$, where $K$ denotes the number of neighbors used in the N2N attention module, and evaluate its inference performance under different $K$ values on both LSFOdyssey and Dynamic Replica. As shown in Table 7, the performance of TAPIP3D improves as $K$ increases, though the gain saturates when $K$ increases from 32 to 48 on LSFOdyssey. The computational complexity of the N2N attention module, given by $O(TNK^2D + TNKD^2)$, makes it a major bottleneck in our model, where $T$ is the number of frames, $N$ is the number of tracked points, and $D$ is the feature dimension. We set $K = 32$ as a trade-off between performance and efficiency.

Table 7: Ablation on Neighborhood Size $K$

| Neighborhood Size $K$ | LSFOdyssey | | | Dynamic Replica | | |
|---|---|---|---|---|---|---|
| | $AJ_{3D} \uparrow$ | $APD_{3D} \uparrow$ | $OA_{3D} \uparrow$ | $AJ_{3D} \uparrow$ | $APD_{3D} \uparrow$ | $OA_{3D} \uparrow$ |
| $K = 16$ | 67.9 | 84.5 | 83.6 | 52.3 | 68.8 | 83.2 |
| $K = 24$ | 71.1 | **85.8** | 85.9 | 55.5 | 73.0 | 84.8 |
| $K = 48$ | **72.6** | 85.3 | **88.6** | **57.3** | **74.3** | **85.8** |
| $K = 32$ | 72.2 | **85.8** | 86.9 | 55.5 | 72.8 | 85.3 |

### E.2  Robustness to Image Blur

Real-world images of dynamic objects often exhibit blurriness, especially under low resolution and fast motion. To assess our model's robustness to such degradation, we evaluate TAPIP3D-world + MegaSaM on the LSFOdyssey dataset using synthetic Gaussian blur of varying strength. For each frame, the input image is convolved with a Gaussian kernel of standard deviation $\sigma \in \{0.0, 0.5, 1.0, 2.0\}$. Since image blur affects the reliability of depth estimation, we re-estimate the depth maps of blurred images using MegaSaM.

As shown in Table 8, our model consistently achieves higher 3D accuracy than the baseline DELTA across all blur levels. We observe that even under heavy blur, the predicted 3D points remain well aligned with scene geometry, although the depth estimated by MegaSaM becomes less consistent across frames.

Table 8: Effect of Gaussian blur on LSFOdyssey. Blurred images are used for both TAPIP3D and DELTA, and corresponding depths are re-estimated using MegaSaM.

| $\beta$ (px) | TAPIP3D (ours) | | DELTA | |
|---|---|---|---|---|
| | $APD_{3D}$ | $APD_{2D}$ | $APD_{3D}$ | $APD_{2D}$ |
| 0.0 | 29.2 | 87.5 | 28.0 | 90.2 |
| 0.5 | 21.2 | 87.4 | 20.3 | 90.3 |
| 1.0 | 20.6 | 86.6 | 19.8 | 90.3 |
| 2.0 | 18.1 | 86.0 | 17.5 | 90.2 |

### E.3 Robustness to Camera Pose Errors

To evaluate our model's robustness against inaccurate camera poses, we synthetically perturb the ground-truth camera orientations. For each frame, we apply a rotation around a random axis, where the rotation angle is sampled from a zero-mean Gaussian distribution with a standard deviation of $\sigma \in \{0.5°, 1.0°, 2.0°\}$.

As reported in Table 9, our model, TAPIP3D-world, demonstrates considerable resilience to these perturbations. The performance degrades gracefully as the noise level increases, and our model consistently outperforms the DELTA baseline even under the perturbation of $\sigma = 2.0°$.

Table 9: Impact of camera pose perturbations. We add Gaussian angular noise with standard deviation $\sigma$ to the ground-truth camera orientations and report results on the LSFOdyssey dataset.

| $\sigma$ (deg) | Method | $AJ_{3D}$ | $APD_{3D}$ | OA |
|---|---|---|---|---|
| N/A | DELTA | 37.1 | 50.1 | 82.3 |
| N/A | TAPIP3D-camera | 68.3 | 83.2 | 86.2 |
| 0.0 | TAPIP3D-world | 72.2 | 85.8 | 86.9 |
| 0.5 | TAPIP3D-world | 70.4 | 84.6 | 86.8 |
| 1.0 | TAPIP3D-world | 67.8 | 82.7 | 86.4 |
| 2.0 | TAPIP3D-world | 64.9 | 80.3 | 85.7 |

## F  Evaluation with Sparse Depth

In some real-world scenarios, we only have access to sparse depth measurements (e.g., from LiDAR sensors) instead of dense depth maps. To assess the usefulness of our approach under such conditions, we construct a benchmark consisting of 50 sequences from the Waymo Open Dataset [37] (a subset of TAPVid-3D), paired with depth map annotations obtained by densifying sparse LiDAR measurements using a simple nearest-neighbor interpolation, following the approach in DriveTrack [1].

We report the performance of **TAPIP3D-camera**, **DELTA**, and **CoTracker3** on this benchmark in Table 10. TAPIP3D-camera achieves the best 3D accuracy among the compared methods. These results demonstrate that our method is highly usable and competitive when only sparse depths are available.

Table 10: Evaluation with sparse depth on a subset of the Waymo Open Dataset [37]. Depth maps are obtained by densifying sparse LiDAR signals via nearest-neighbor interpolation.

| Method | $AJ_{3D}$ | $APD_{3D}$ | $AJ_{2D}$ | $APD_{2D}$ |
|---|---|---|---|---|
| CoTracker3 | 16.5 | 24.1 | 63.7 | 78.8 |
| DELTA | 21.0 | 30.8 | **71.4** | **85.0** |
| TAPIP3D-camera | **21.8** | **31.6** | 67.7 | 82.2 |

# G  TAP-Vid Evaluation

For reference, we also evaluated TAPIP3D-world on TAP-Vid-DAVIS [7], a 2D point tracking benchmark. TAPIP3D-world achieves an AJ of 58.9, an APD of 71.2, and an OA of 89.6, which are lower than those of recent state-of-the-art 2D trackers.

We note that TAPIP3D is not designed to achieve state-of-the-art 2D tracking performance, and the 2D tracks obtained by projecting 3D trajectories to 2D cannot reach sub-pixel accuracy in monocular settings. The performance is further affected by MegaSaM's failures on some of sequences, which lead to severely noisy depth estimates and unstable point clouds.

# H  3D Track Visualization

For extensive 3D visualizations of our results, please refer to the accompanying project page, where we compare our method with the baselines SpaTracker [48], CoTracker3 [19] and DELTA [30].

