# OpenReview forum: "TAPIP3D: Tracking Any Point in Persistent 3D Geometry"
_NeurIPS.cc/2025/Conference — NeurIPS 2025 poster_

### Official Review · Reviewer_6eUD · 2025-06-16

**Clarity:** 3
**Significance:** 2
**Originality:** 3
**Rating:** 4
**Confidence:** 5

**Summary:**

This paper introduces TAPIP3D, a method for long-term point tracking in 3D by explicitly operating in a world-centric coordinate frame. Unlike prior approaches that append depth to 2D pixel coordinates (UVD), TAPIP3D lifts features directly into (XYZ) using estimated depth and camera poses. These features are organized into spatio-temporal 3D feature clouds, with camera motion compensated via extrinsics. To model correspondence across time, the authors propose a 3D Neighborhood-to-Neighborhood (N2N) Attention module, where query features attend to local neighborhoods in 3D, avoiding 2D grid biases. The model achieves strong performance on standard 3D tracking benchmarks.

**Questions:**

In addition to the concerns mentioned under weaknesses, the reviewer has the following additional questions:
- How does the model behave when camera extrinsics are inaccurate? Is there any fallback or degradation analysis?
- How critical is the CoTracker initialization? What happens with random initialization?
- What if a different geometry estimator, such as MonST3R [4], was used instead of MegaSaM? An ablation would be useful here.

---
## References
[4] Zhang et al., Monst3r: A simple approach for estimating geometry in the presence of motion, ICLR 2025

**Ethical Concerns:**

["NO or VERY MINOR ethics concerns only"]

**Final Justification:**

The rebuttal addresses several important concerns, including the impact of different camera pose estimation methods and runtime analysis (as detailed in the discussion below). However, the conceptual advantages of 3D tracking over strong 2D baselines remain somewhat **underexplored**. It is encouraged that the final version of the paper further elaborates on this aspect, as understanding the trade-offs between 2D and 3D tracking is crucial for practical applications.

Given the productive discussion during the rebuttal period and the novel contributions presented in the paper, the score is increased, and acceptance is recommended.

**Limitations:**

yes

**Paper Formatting Concerns:**

No concerns.

**Quality:**

3

**Strengths And Weaknesses:**

## Strengths
- Using camera extrinsics to eliminate camera motion is a simple yet effective idea for robust 3D tracking.
- The Neighborhood-to-Neighborhood Attention module is a meaningful architectural contribution, allowing the model to reason over 3D geometry directly rather than relying on fixed 2D neighborhoods or triplane projections.
- The paper is clearly written, and figures are informative.
- Ablation studies are thorough and highlight where the improvements come from, especially in high-motion scenes.

---

## Weaknesses
- It is important to demonstrate the benefit of 3D tracking over 2D. The paper does not compare against strong 2D trackers like Track-On [1], CoTracker3 [2], or TAPNext [3], which is especially relevant in setups like Dynamic Replica where GT camera parameters are available. Comparisons on TAP-Vid 2D would also be informative.
- Despite the architectural similarity to CoTracker3, the occlusion accuracy is considerably lower (Table 1). Some explanation is needed.
- The paper claims 10 FPS runtime, but it is unclear whether this includes camera pose estimation. A breakdown of latency and memory use for each module would give a clearer picture of efficiency.

---

## Overall
The paper shows clear improvements over existing 3D trackers, with a simple but effective idea of reasoning in world space. However, comparisons against strong 2D baselines are missing, which limits how broadly the results can be interpreted.

---

## References
[1] Aydemir et al., Track-On: Transformer-based Online Point Tracking with Memory, ICLR 2025

[2] Karaev et al., CoTracker3: Simpler and Better Point Tracking by Pseudo-Labelling Real Videos, arXiv 2025

[3] Zholus et al., TAPNext: Tracking Any Point (TAP) as Next Token Prediction, arXiv 2025

---

> ### Author Rebuttal · Authors · 2025-07-31
>
> ### Response to Reviewer 6eUD
> **1. Lack of Comparisons with 2D Trackers (Track-On, CoTracker3, or BootsTAPNext) on Dynamic Replica**
>
>
> Our submission already includes quantitative comparisons with CoTracker3, including on the Dynamic Replica dataset, as in Table 2 of the paper. Per your request, we now include additional experiments with BootsTAPNext [a] on both Dynamic Replica and LSFOdyssey datasets as below. While BootsTapNext exhibits slightly better performance than CoTracker3 on certain metrics, our method still demonstrates superior results in both 2D and 3D metrics when ground truth depth is available.
>
>
>
> | Dataset           | Method              | AJ-2D | APD-2D | AJ-3D | APD-3D |
> |-------------------|---------------------|-------|--------|-------|--------|
> | LSFOdysey         | CoTracker3 + GT     | 75.0  | 88.0   | 28.4  | 35.0   |
> |                   | BootsTAPNext + GT   | 75.9  | 89.5   | 24.9  | 30.9   |
> |                   | TAPIP-world + GT    | **78.5**  | **92.8**   | **72.2**  | **85.8**   |
> | Dynamic Replica   | CoTracker3 + GT     | 61.4  | 80.0   | 27.4  | 38.0   |
> |                   | BootsTAPNext + GT   | 38.7  | 53.7   | 15.8  | 22.8   |
> |                   | TAPIP-world + GT    | **66.2**  | **85.7**   | **55.5**  | **72.8**   |
>
> > Note: BootsTAPNext's performance on Dynamic Replica is inferior in the table. We consider this as normal, as mentioned in Section 4.3 of [a]: "We observe significant failure in long-term point tracking over the full length of videos greater than 150 frames.", while sequences in Dynamic Replica have 300 frames.
>
> However, we wish to clarify that our goal is not 2D tracking. Accordingly, we focus evaluations on datasets that provide 3D point-track annotations, which allows us to compute 3D point tracking metrics. Note that [a] is public on arXiv only 1 month before the paper submission DDL. We will include more detailed comparison to BootsTAPNext [a] and Track-On in our final version.
>
> [a] TAPNext: Tracking Any Point (TAP) as Next Token Prediction[J]. arXiv preprint arXiv:2504.05579, 2025.
>
> **2. Lower occlusion accuracy**
>
> We acknowledge this limitation, which stems from TAPIP3D’s 3D-centric design. The model operates entirely in XYZ coordinates using KNN-based attention, without explicit knowledge of pixel-space proximity or depth ordering. As a result, occlusion reasoning relies solely on neighbor features and their spatial offsets, making it more challenging, especially when depth reconstruction is unreliable.
>
> This issue is particularly noticeable in the DriveTrack dataset, where occlusions between distant vehicles lead to significant depth estimation errors. In some cases, depth of remote objects becomes visually fused due to the low resolution of the underlying depth model, which results in misleading geometry and unreliable neighbor associations. Incorporating auxiliary 2D cues, such as depth-difference maps used in SpaTracker and DELTA, could help address this, which we leave for future work.
>
> **3. Clarification on Latency and Memory Use**
>
> We appreciate the reviewer’s request for a more detailed runtime and memory analysis. The reported 10 FPS refers to the tracking stage, assuming that depth and camera poses are available. It does not include the runtime of depth or pose estimation, as these are obtained from external sensors or off-the-shelf models (e.g., MegaSaM [9], CUT3R [10]) and thus can  vary dramatically. Importantly, our model is not trained with any specific estimation method and can be flexibly combined with any future advancements in depth or pose estimation.
>
> Below, we provide a breakdown of latency during one pass of forward tracking on a sequence from the Dex-YCB dataset on a single L40S GPU. The forward pass consumes a total peak memory usage of 3.6GB, while feature encoding uses up to 2.8GB.
>
>
> | Module  | Feature Extraction |  Correlation Feature Encoding (KNN & N2N Attention) | Updating Transformer | Overall |
> |-----------------------|-----------------|-----------|------------------------|--------------|
> | Latency (s)           | 0.11          |    6.04    | 0.25                     | 6.4          |
>
>
> We will include these in the revised manuscript.
>
>
> **4. On Question 2: Model Behavior and Fallback under Inaccurate Camera Extrinsics**
>
> Although TAPIP3D relies on estimated camera poses to transform points into a shared world frame, our training augmentations enable the model to generalize well under smooth camera motion. As a result, the model retains a degree of robustness even when the poses are inaccurate. As shown in Tables 1 and 2 of the paper, TAPIP3D-world and TAPIP3D-camera share the same model but use different pose inputs. TAPIP3D-camera effectively serves as a lower bound on the robustness of our approach, achieving competitive results even when camera extrinsics are highly unreliable, in which case the user can simply specify identity matrices as extrinsics.
>
> **5. Dependence on CoTracker Initialization**
>
> We incorporated CoTracker3 initialization early in the project and observed that it accelerates convergence; however, it should have minimal impact on the final performance after full training. We will include an ablation study in the revised paper to substantiate this observation.
>
> **6. Ablation on Depth Estimator Choice**
>
> We would like to clarify that our model is not specifically trained for any particular geometry estimation method. For evaluation, we selected MegaSAM due to its superior geometric accuracy and robust camera pose estimation.
>
> Here, we additionally include results using UniDepth [a], a weaker depth estimation model with lower geometric and pose accuracy than MegaSAM. While the overall performance drops, our model still produces reasonable results.
>
>
> | Method                             | AJ-3D | APD-3D | AJ-2D | APD-2D | OA   |
> |----------------------------------|-------|--------|-------|--------|------|
> | **Using UniDepth**                |       |        |       |        |      |
> | CoTracker3 + UniDepth             | 19.2  | 29.9   | **61.4** | **80.0** | **86.2** |
> | SpatialTracker + UniDepth         | 7.6   | 13.5   | 53.8  | 72.6   | 81.3 |
> | DELTA + UniDepth                  | 18.8  | 28.4   | 61.0  | 77.5   | 81.0 |
> | **TAPIP3D-camera** + UniDepth | **20.6** | **30.3** | 55.3  | 77.1   | 80.2 |
> |                                  |       |        |       |        |      |
> | **Using MegaSaM**                 |       |        |       |        |      |
> | CoTracker3 + MegaSaM              | 20.0  | 30.2   | 61.4  | 80.0   | **86.2** |
> | SpatialTracker + MegaSaM          | 6.6   | 12.0   | 54.1  | 72.8   | 81.7 |
> | DELTA + MegaSaM                  | 18.0  | 27.2   | 61.3  | 77.3   | 81.0 |
> | **TAPIP3D-camera** + MegaSaM | **20.6** | **30.3** | 55.3  | 77.1   | 80.2 |
> | **TAPIP3D-world** + MegaSaM  | **20.6** | 30.2   | **56.1** | **77.7** | 78.6 |
> |                                  |       |        |       |        |      |
> | **Using GT Camera Poses and Depth**                 |       |        |       |        |      |
> | CoTracker3 + GT             | 27.4  | 38.0   | 61.4  | 80.0   | **86.2** |
> | SpatialTracker + GT        | 7.6   | 13.6   | 56.4  | 74.7   | 83.8 |
> | DELTA + GT                   | 27.4  | 37.7   | 65.6  | 80.5   | 83.8 |
> | **TAPIP3D-camera** + GT | 53.7  | 70.8   | 64.6  | 84.7   | 84.7 |
> | **TAPIP3D-world** + GT | **55.5** | **72.8** | **66.2** | **85.7** | 85.3 |
>
> [a] UniDepth: Universal Monocular Metric Depth Estimation. CVPR, 2024.

---

> > ### Comment · Reviewer_6eUD · 2025-08-03
> >
> > The additional clarifications and experiments are appreciated, and all reviews and responses have been carefully considered.
> > * While the primary scope of this work is 3D tracking, the suggestion to compare against strong 2D trackers was intended not to emphasize outperforming them, but to better understand the conceptual and empirical trade-offs between 2D and 3D tracking formulations. In scenarios where 3D cues are not strictly necessary, it remains unclear whether 3D reasoning provides consistent benefits. Existing works like SpatialTracker also operate in 3D but include 2D benchmark comparisons for broader perspective. Including TAP-Vid results would serve as a valuable reference point, especially since it remains a widely used benchmark.
> > * The clarification regarding the CoTracker3 baseline is helpful. The original understanding was that CoTracker3 had been retrained in 3D using depth-augmented inputs, whereas in fact the method uses 2D predictions projected into 3D. This distinction is now clear.
> > * The inclusion of new experiments with alternative depth estimators (UniDepth) is appreciated and demonstrates some degree of robustness across depth sources. However, as also noted by *Reviewer pGLy* and in the original review, the sensitivity of the model to camera pose estimation remains a crucial factor for assessing practical utility. A more complete analysis, such as runtime and accuracy breakdowns when using different pose estimation methods, would help contextualize the feasibility and generalizability of the pipeline in real-world applications.

---

> ### Author Response · Authors · 2025-08-08
> **Response to Reviewer 6eUD**
>
> Thank you for the thoughtful follow-up. We address your remaining concerns as below:
>
> ### **1. Sensitivity to Camera Pose Estimation**
>
> We acknowledge that demonstrating robustness across different depth estimators alone could be insufficient. We provide two complementary experiments to assess pose estimation sensitivity, where our TAPIP3D is robust to varing pose estimation methods and independent angular camera noise:
>
> **Experiment 1: Alternative Pose Estimation Method**
> To provide a broader comparison beyond the MegaSAM + MoGe pipeline, we evaluate our method on LSFOdyssey using ground truth depth combined with camera poses estimated by various methods.
>
> | Method                    | AJ-3D | APD-3D | OA|
> |---------------------------|-------|--------|-------|
> | TAPIP3D-camera       | 68.3    | 83.2    | 86.2  |
> | TAPIP3D-world + GT poses     | 72.2    | 85.8    | 86.9  |
> | TAPIP3D-world + MegaSaM poses     | 71.9   | 86.5    | 87.2   |
> | TAPIP3D-world + CUT3R poses     | 71.6   | 85.6    | 86.9   |
>
> Note that TAPIP3D-camera represents our method's performance when pose information is completely unavailable, where we simply pass identity matrices as extrinsics.
>
> [a] Continuous 3D Perception Model with Persistent State. CVPR, 2025.
>
> **Experiment 2: Injecting Angular Noise to Camera Poses**
> We inject angular noise into GT camera poses on LSFOdyssey. Each frame's camera orientation is independently perturbed by rotating around a random axis, creating significant geometric inconsistencies across frames. Even with large angular noise like 2.0°, our method still significantly outperforms the baseline method DELTA.
>
>
> | σ (°)     | Method                     | AJ-3D | APD-3D | OA   |
> |-----------|----------------------------|--------|--------|------|
> | N/A       | DELTA (ICLR'25) |   37.1  | 50.1    | 82.3  |
> | N/A       | TAPIP3D-camera | 68.3    | 83.2    | 86.2  |
> | 0.0       | TAPIP3D-world      | 72.2    | 85.8    | 86.9  |
> | 0.5       | TAPIP3D-world      | 70.4    | 84.6    | 86.8  |
> | 1.0       | TAPIP3D-world      | 67.8    | 82.7    | 86.4  |
> | 2.0       | TAPIP3D-world     | 64.9    | 80.3    | 85.7  |
>
> *σ represents the standard deviation of the angular noise
>
>
> ### **2. TAP-Vid Evaluation**
>
> Following your suggestion, we conducted an evaluation on the TAP-Vid-DAVIS benchmark on the 2D tracking metrics (APD-2D), where CoTracker3 achieves 77.0, while our TAPIP3D-world reaches 71.2. The performance gap is primarily due to MegaSAM's complete failure in partial video scenarios (4/30), resulting in severely noisy depth estimates and unstable point clouds. If we removed this 4 MeGa-SaM failing DAVIS video, the 2D point tracking of CoTracker3 is 77.1 while ours is 73.4. We discussed these in the limitations section and the supplemental video, and we will include more detailed per-video experimental analysis in the final version of the paper.
>
> We would also like to clarify that TAPIP3D is designed for two primary application scenarios and 3D point tracking:
> 1) **RGB-D settings**, using sensor depth, such as in robotics or autonomous driving (Table 3 of the paper, TAPIP3D achived significant boost);
> 2) **Monocular videos**, where the accuracy and robustness of MegaSAM-like depth and pose estimators are rapidly improving.
>
> As noted by Reviewer Sbrq:
> > “I believe the field of camera pose and depth estimation is growing surprisingly fast, supported by recent works such as VGGT [1], CUT3R [2], and pi-3 [3]. I'm confident that these concerns will diminish in the near future, making this work even more important as TAPIP-3D stands to benefit greatly from those advancements.”
>
> We would like to emphasize that this comparison is meant to highlight trade-offs and limitations, rather than to claim TAPIP3D's superiority over established 2D tracking methods.
>
> ###  **3. Additional Runtime Analysis**
>
> We provide latency breakdown for two geometry estimation pipelines on a sequence from Dex-YCB using a single L40S GPU:
>
> | Component                  | Latency (s) |
> |----------------------------|-------------------|
> | Given Camera Poses with Sensor Depth (Robotics/Driving Scenario)  | 0.       |
> | Geometry estimation (using MegaSAM + MoGe)  | 121.1               |
> | Geometry estimation (using CUT3R)          | 6.54               |
> | Tracking (Our model)       | 6.4               |
> | **Total (Given camera pose and depth)** | 6.4               |
> | **Total (MegaSAM + MoGe)** | 127.5               |
> | **Total (CUT3R)**          | 12.94               |

---

> > ### Comment · Reviewer_6eUD · 2025-08-08
> >
> > The additional explanations and experiments provided in the rebuttal are appreciated and clarify several key points.
> > * The experiments with alternative pose estimation methods are informative. They highlight that camera pose accuracy plays a critical role when combined with estimated depth. Notably, it is interesting to observe that using less accurate pose estimates (MegaSAM) can sometimes lead to better performance than ground-truth poses. While an explanation for this behavior would be valuable, given the limited time remaining in the discussion period, the reviewer does not request a detailed response but strongly encourages the authors to address this in the final version of the paper.
> > * Regarding the TAP-Vid-DAVIS evaluation, the performance gap remains substantial even after noisy depth estimates are removed. This is somewhat surprising given architectural similarities, i.e. iterative update transformers. It is recommended that the authors include comparisons to additional 2D tracking models to provide a broader context.
> > * The runtime breakdown is helpful and shows that camera pose estimation constitutes the main computational bottleneck. The observation that switching to a more efficient pose estimator (CUT3R vs. MegaSAM + MoGe) leads to significant speedups is encouraging.
> >
> > While many of the raised concerns have been addressed, it is recommended that the authors incorporate the above suggestions to provide a more comprehensive perspective in the final version.

---

### Official Review · Reviewer_pGLy · 2025-06-26

**Clarity:** 3
**Significance:** 2
**Originality:** 3
**Rating:** 4
**Confidence:** 4

**Summary:**

The paper proposes a new approach for 3D point tracking using RGB or RGB-D data. It uses the off-the-shelf method MegaSAM to recover camera motions and depth maps.
Then, the per-frame point cloud features are lifted and transformed to the world frame for better stability against camera motion.
The 3D points are tracked by a 3D Neighborhood-to-Neighborhood attention mechanism.
Experiments are conducted on TAPVid3D, DexYCB-Pt, LSFOdyssey, and DynamicReplica datasets and compared with camera frame 3D tracking methods, SpatialTracker, DELTA, and 2D tracking method CoTracker3.

**Questions:**

(1) The average performance of DELTA in Table 1 is 22.3, which is significantly lower than Cotrackerv3 and the proposed approach. However, its results on separately reported datasets are pretty competitive. The reviewer is curious about the reason for this performance gap. Is it a mistake?

(2) In Table 2' GT depth version, the performance of the proposed approach (camera frame) is slightly (<5%) outperformed by the strong 2D tracker CoTracker3 on 2D tracking metrics, but the performance of CoTracker3 is significantly worse on 3D metrics (>40%). Since both of these approaches use GT depth maps, the reviewer is curious about the reason for this performance gap.

**Ethical Concerns:**

["NO or VERY MINOR ethics concerns only"]

**Final Justification:**

The authors have addressed the major concerns of the reviewer. The reviewer is glad to raise the score.

**Limitations:**

Yes

**Quality:**

3

**Strengths And Weaknesses:**

# Strengths

(1) The experiments are sufficient, which are conducted on several large datasets with thousands of sequences and cover various scenarios.
(2) The strategy of using estimated camera motions to convert each frame's point cloud to the world frame for tracking is reasonable and effective, which can reduce the influence of camera motion.
(3) The overall writing is clear, and the method is well explained.

# Weaknesses

(1) The reviewer's major concern is the advantage of the proposed 3D tracking architecture over existing approaches (DELTA and directly lifting 2D tracked points of CoTracker3 to 3D). Firstly, the idea of transforming the point cloud to the world frame for tracking is straightforward as long as the camera motion is estimated. Ideally, this approach can be applied to other methods, where a 3D tracker may need retraining on such a type of world frame data. Without transforming to the world frame strategy, the reviewer finds that the performance gain of the proposed approach over strong baseline methods DELTA and CoTracker3 is very marginal, as shown in Table 1 and Table 2 (MegaSAM version)
Specifically, the performance improvement in Table 1 is generally around 1%, which is not significant.
In Table 2's MegaSAM version, the performance improvement over state-of-the-art CoTracker3 on 3D tracking metrics is around 1% and 0.5% on LSFOdyssey and Dynamic Replica datasets, respectively.

(2) Since the approach performs tracking on point clouds, its by nature sensitive to the quality of depth maps and camera motions: In Table 2's MegaSAM version with depth noise, the performance of the proposed method on 2D tracking metrics is significantly lower than 3D tracker DELTA using same depth maps on several metrcis.

(3) The sensitivity of TAPIP3D-world version on camera motion estimation is not well studied and discussed.

---

> ### Author Rebuttal · Authors · 2025-07-31
>
> ### Response to Reviewer pGLy
> **1. On the Contribution and Necessity of TAPIP3D’s Architecture**
>
> We agree with the reviewer that the performance improvements of our camera space approach using estimated depth may appear modest in Table 1. However, we respectfully disagree with the suggestion that operating in a stabilized world frame is a straightforward extension of existing baselines or that our architectural contributions are incremental.
>
> Most prior methods, including DELTA and CoTracker3, are fundamentally designed around 2D-centric architectures. They operate in UVD space, which is 2D coordinates with depth appended, and rely on grid-based correlation volumes defined in pixel space. These components are not naturally extendable to the unordered and irregular structure of world-frame 3D data, where no global 2D grid exists.
>
> In TAPIP3D, we address this challenge with our 3D Neighborhood-to-Neighborhood (N2N) attention, which enables local feature matching and aggregation directly in a shared world coordinate space. To the best of our knowledge, there is no simple way to adapt 2D-based models like DELTA or CoTracker3 to this setting without significant architectural changes.
>
> We also note that the performance gap becomes substantially larger when higher-quality geometric input is available. In Table 2 (GT depth), TAPIP3D outperforms prior methods by over 30% on the 3D APD metric. Similarly, Table 3 (DexYCB) shows that our architecture achieves significantly better performance than prior baselines when using sensor depth, which is often available in robotic scenario. This demonstrates that TAPIP3D is well-suited to leverage accurate geometric inputs, which is an increasingly important advantage as 3D foundation models for depth and pose estimation continue to improve.
>
> **2. On the Sensitivity of the method to Depth Input**
>
> While the reviewer notes that our 2D tracking performance using MegaSaM depth appears weaker, our primary contribution lies in robust 3D tracking. As shown in Table 2, TAPIP3D consistently outperforms all baselines in 3D metrics (AJ_3D and APD_3D) under both MegaSaM and GT depth, demonstrating the method’s robustness to depth quality. Moreover, the 2D performance of TAPIP3D improves significantly when higher-quality depth is available (e.g., GT depth), indicating that the 2D results are naturally coupled with depth accuracy rather than a limitation of the model itself.
>
> In many real-world scenarios such as robotics or sensor-based capture, accurate depth maps are readily available from RGB-D cameras. As evidenced by the sensor depth results in Table 3, TAPIP3D achieves state-of-the-art 3D tracking in such settings, which highlights a key advantage of our approach. To further validate this, we provide evaluation results on a subset of DriveTrack [1] using sparse LiDAR depth, densified via a simple nearest neighbor approach. Despite the presence of noise and significant discontinuities in the unprojected geometry, our model achieves robust 2D and 3D tracking:
>
> | Methods               | AJ-2D | APD-2D | AJ-3D | APD-3D |
> |-----------------------|-------|--------|-------|--------|
> | Cotracker3 + LiDAR       |  63.7     |   78.8     |   16.5     |   24.1    |
> | DELTA + LiDAR       |  **71.4**     |     **85.0**   |   21.0      | 30.8     |
> | TAPIP3D-camera + LiDAR   |  67.7     |     82.2  |   **21.8**    |   **31.6**     |
>
>
> We additionally include results using UniDepth [a], a weaker depth estimation model with lower geometric and pose accuracy than MegaSAM. While the overall performance drops, our model still produces reasonable results.
>
>
> | Method                             | AJ-3D | APD-3D | AJ-2D | APD-2D | OA   |
> |----------------------------------|-------|--------|-------|--------|------|
> | **Using UniDepth**                |       |        |       |        |      |
> | CoTracker3 + UniDepth             | 19.2  | 29.9   | **61.4** | **80.0** | **86.2** |
> | SpatialTracker + UniDepth         | 7.6   | 13.5   | 53.8  | 72.6   | 81.3 |
> | DELTA + UniDepth                  | 18.8  | 28.4   | 61.0  | 77.5   | 81.0 |
> | **TAPIP3D-camera** + UniDepth | **20.6** | **30.3** | 55.3  | 77.1   | 80.2 |
> |                                  |       |        |       |        |      |
> | **Using MegaSaM**                 |       |        |       |        |      |
> | CoTracker3 + MegaSaM              | 20.0  | 30.2   | 61.4  | 80.0   | **86.2** |
> | SpatialTracker + MegaSaM          | 6.6   | 12.0   | 54.1  | 72.8   | 81.7 |
> | DELTA + MegaSaM                  | 18.0  | 27.2   | 61.3  | 77.3   | 81.0 |
> | **TAPIP3D-camera** + MegaSaM | **20.6** | **30.3** | 55.3  | 77.1   | 80.2 |
> | **TAPIP3D-world** + MegaSaM  | **20.6** | 30.2   | **56.1** | **77.7** | 78.6 |
> |                                  |       |        |       |        |      |
> | **Using GT Camera Poses and Depth**                 |       |        |       |        |      |
> | CoTracker3 + GT             | 27.4  | 38.0   | 61.4  | 80.0   | **86.2** |
> | SpatialTracker + GT        | 7.6   | 13.6   | 56.4  | 74.7   | 83.8 |
> | DELTA + GT                   | 27.4  | 37.7   | 65.6  | 80.5   | 83.8 |
> | **TAPIP3D-camera** + GT | 53.7  | 70.8   | 64.6  | 84.7   | 84.7 |
> | **TAPIP3D-world** + GT | **55.5** | **72.8** | **66.2** | **85.7** | 85.3 |
>
> [a] UniDepth: Universal Monocular Metric Depth Estimation. CVPR, 2024.
>
> **3. On the Sensitivity of the method to Camera Pose Estimation**
>
> Although TAPIP3D relies on estimated camera poses to transform points into a shared world frame, our training augmentations enable the model to generalize well under smooth camera motion. As a result, the model retains a degree of robustness even when the poses are inaccurate. As shown in Tables 1 and 2 of the paper, TAPIP3D-world and TAPIP3D-camera share the same model but use different pose inputs. TAPIP3D-camera effectively serves as a lower bound on the robustness of our approach, achieving competitive results even when camera extrinsics are highly unreliable, in which case the user can simply specify identity matrices as extrinsics.
>
> **4. Evaluation Clarifications**
>
> **3.1 Discrepancy on the reported  Average Performance of DELTA in Table 1**
>
> This is a typo, thank you for pointing this out! We incorrectly computed the DELTA average in Table 1. The per-dataset values are correct, and we have updated this average to its correct value from 22.3 to 26.3 in the manuscript. Our TAPIP3D-world still outperforms DELTA for 1.1 APD$_{3D}$.
>
> **3.2 Why the proposed method has a much better 3D tracking performance that Cotracker3 despite worse 2D tracking performance, since both methods use GT depth?**
>
> Thank you for the question. We would like to clarify a possible misunderstanding. In Table 2 (GT depth version), **TAPIP3D-camera actually outperforms CoTracker3 on both 2D and 3D tracking metrics**, not just in 3D.
> - **LSFOdyssey**: *APD-3D*: 83.2 vs. 35.0; *APD-2D*: 91.2 vs. 88.0
> - **Dynamic Replica**: *APD-3D*: 70.8 vs. 38.0; *APD-2D*: 84.7 vs. 80.0
>
> The improvement appears numerically larger for 3D metrics because when using the same depth map, even small 2D pixel offsets—particularly for points near object boundaries—can lead to significant depth variation (e.g., a point jumping from foreground to background). This amplifies the effect in 3D space, making gains in 2D accuracy translate into larger improvements in 3D error.
>
> [1] DriveTrack: A Benchmark for Long-Range Point Tracking in Real-World Videos. CVPR, 2024.

---

> > ### Comment · Reviewer_pGLy · 2025-08-07
> >
> > The reviewer appreciates the authors' efforts in rebuttal and has carefully read other reviewers' comments. The reviewer has some remaining comments after the rebuttal.
> >
> > 1. The reviewer understands the comment that the previous 3D tracking approach, DELTA, cannot handle non-grid point clouds in the world frame due to the need to concatenate depth maps, whereas the proposed architecture can handle this type of data. It seems that SpatialTracker's architecture can be directly adapted to world-coordinate point clouds.
> >
> > 2. The reviewer is curious about why there is inconsistent performance improvement between 3D tracking metrics and 2D tracking metrics, e.g., TAPIP3D-camera + LiDAR is marginally (around 0.8%) better than DELTA + LiDAR on 3D metrics, but worse (around 4%) on 2D metrics on the Drivetrack dataset. Similar observations also hold for UniDepth and MegaSAM experiments.

---

> > > ### Author Response · Authors · 2025-08-08
> > >
> > > We thank the reviewer for the thoughtful comments and the opportunity to clarify our contribution. While it may seem that SpatialTracker’s architecture could be directly adapted to handle world-coordinate point clouds, TAPIP3D is the first to explicitly propose and validate a model that operates natively on unstructured 3D point clouds in the world frame (XYZ), rather than relying on image-aligned grids or concatenated depth maps. Prior works such as DELTA and SpatialTracker are fundamentally limited by their reliance on structured, camera-centric representations. Our architecture is designed to overcome this limitation, enabling tracking on sparse, non-grid, and sampled 3D points—an important step toward real-world applicability in robotics and vision. We will revise the manuscript to more clearly highlight the distinction between adapting grid-based methods to point clouds versus designing architectures that natively operate in the world-coordinate space.
> > >
> > > Regarding the second question, the discrepancy between 2D and 3D metrics primarily stems from differences in how prediction errors are distributed in 3D space. For UVD-based methods, pixel coordinates (UV) and depth (D) are decoupled, meaning that even when 3D predictions are poor, the model can still predict correct pixel coordinates by leveraging pixel-space correlations. This causes errors to be distributed primarily along the D-axis (i.e., the camera ray direction). In contrast, our model performs tracking directly in XYZ space and discards 2D grid-based correlations. While our approach achieves better 3D accuracy, decoupling pixel coordinates from depth becomes non-trivial in this representation. When 3D tracking becomes challenging, errors distribute isotropically throughout the 3D space, which then translate into errors when projected back to 2D. This explains why our model sometimes demonstrates superior 3D accuracy but lower 2D accuracy, especially when high-quality depth information is unavailable.

---

### Official Review · Reviewer_Sbrq · 2025-07-01

**Clarity:** 4
**Significance:** 4
**Originality:** 4
**Rating:** 6
**Confidence:** 5

**Summary:**

This paper addresses the problem of 3D point tracking. It begins by extracting depth and camera parameters from MegaSAM, with the flexibility to incorporate higher-quality depth data if available. Subsequently, it extracts hierarchical image features using CoTracker's feature backbone. These image features are then concatenated with point maps derived from MegaSAM. The core of the model involves a transformer-based mechanism that performs "neighbors-to-neighbors" attention. This approach can be conceptually understood as a 3D extension of the 4D local correlation found in LocoTrack, or, from another perspective, as a local-to-local point-cloud registration. A crucial distinction is that point cloud matching is performed in world space rather than camera space, a formulation that simplifies trajectory complexity, as illustrated in Figure 1.

**Questions:**

I have a question regarding the substantial performance gap between TAPIP-3D and its competitors in LSFOdyssey; further explanation would be beneficial. Additionally, it's somewhat perplexing to observe that SpatialTracker's performance in LSFOdyssey drops when using ground truth depth compared to MegaSAM predictions.

**Ethical Concerns:**

["NO or VERY MINOR ethics concerns only"]

**Final Justification:**

I've read the authors' rebuttal and all the reviews carefully.

While there are concerns about the model's robustness with noisy depth or pose, and its 2D tracking performance, I believe this criticism is mainly based on the assumption of a 2D monocular-video application, which is a problem that existing 2D trackers have already widely addressed. However, this assumption may not be valid when considering the future application of 3D point tracking. Point tracking is already widely used in robotics and could be used more to extract motion from video [1,2,3,4]. In such scenarios, it is easy to obtain near-ground-truth depth from sensors or multi-view setups, which TAPIP-3D can greatly benefit from, as demonstrated in the TAPVid-3D-ADT experiment from the rebuttal. This property would highly beneficial for learning precise manipulation. Considering this trend, I find this paper extremely useful for these applications and believe this model could become a solid standard for them. Therefore, I am raising my score and strongly recommend this paper for an oral or spotlight presentation.

[1] Bharadhwaj et al., "Track2Act: Predicting Point Tracks from Internet Videos enables Generalizable Robot Manipulation", ECCV'24
[2] Kim et al., "UniSkill: Imitating Human Videos via Cross-Embodiment Skill Representations", CoRL'25
[3] Li et al., "HAMSTER: Hierarchical Action Models For Open-World Robot Manipulation", ICLR'25
[4] Levy et al., "P3-PO: Prescriptive Point Priors for Visuo-Spatial Generalization of Robot Policies", ICRA'25

**Limitations:**

yes

**Paper Formatting Concerns:**

No formatting concerns.

**Quality:**

4

**Strengths And Weaknesses:**

**Strength**
- The paper is well-written, well-structured, and easy to understand.

- The proposed idea of matching the 3D structure of a query point to its neighbors in world space, rather than camera space, is well-motivated. I believe this approach will become a standard in future 3D point tracking research.

- An interesting result is that TAPIP-3D can outperform CoTracker3 in 2D tracking, especially when ground truth depth is provided.

**Major Weaknesses**
- As mentioned in the strengths, it's intriguing that TAPIP surpasses CoTracker3 in 2D tracking when ground truth depth is available. I am curious if this performance trend holds true for the TAPIP-3D dataset as well.

- A key suggestion is to evaluate performance using sparse depth. In real-world scenarios, obtaining dense depth from sensors is almost impossible; most setups yield sparse depth. For instance, the Waymo dataset used in DriveTrack includes sparse depth ground truth from LiDAR. It would be valuable to investigate TAPIP-3D's performance with such sparse depth. One option would be to apply depth completion techniques to the sparse depth and then assess performance, which would make the paper more comprehensive and address the practical demand for using TAPIP-3D with real-world sensor data.

**Minor Weaknesses**
- Before Line 143, the notation 'T' is used to denote the number of frames in a video. However, in Lines 143-144 and Line 176, it is used to represent the size of a local window. This dual usage can mislead readers, and I recommend introducing a distinct notation for the local window size.

- It appears the paper exclusively uses APD with a fixed threshold. It would be beneficial to see the depth-adaptive APD, as introduced in TAP-Vid-3D, also evaluated.

- It would be helpful to include an intuition regarding the failure cases of DELTA, as depicted in Figure 4, within the main text. Specifically, why do the predicted points diverge significantly from the MegaSAM depth prediction? Does this stem from depth prediction failures or 2D tracking errors? If it's a depth error, why does such drifting from MegaSAM occur? In my opinion, simply unprojecting accurate 2D tracking with MegaSAM depth should not lead to such errors.

**Overall, I strongly support this paper. I identify two main novelties:**

- **Minor**: The finding that performing 3D point tracking in world space is advantageous.

- **Major**: The technical novelty of establishing correspondence using 3D geometry features. In previous 3D point tracking works such as Seurat (CVPR'25), most showed marginal improvement over baselines that simply unprojected 2D tracking using depth. I believe the primary reason for this limited improvement was an over-reliance on 2D features, which prevented them from fully leveraging the 3D setup. However, TAPIP-3D matches the geometry around one point to other sets of points, similar to how point cloud registration works, thus benefiting from accurate 3D geometry.

These ideas, I believe, will significantly advance the understanding of 3D point tracking task.

---

> ### Author Rebuttal · Authors · 2025-07-31
>
> ### Response to Reviewer Sbrq
>
> Thank you for your encouraging feedback and for recognizing both the conceptual ideas and technical contributions of our work.
>
> **1. Performance Comparison on TAPVid-3D with Ground Truth Depth**
>
> We appreciate the reviewer’s interest in whether TAPIP-3D’s superior 2D performance when using groundtruth depth generalizes to the TAPVid-3D dataset. However, TAPVid-3D does not provide ground truth depth annotations.
>
> To partially address this question, we investigated the Aria Digital Twin (ADT) sub-dataset of TAPVid3D benchmark, where all objects in the 3D scenes are modeled with provided rendered depth. As in the below table, our TAPIP3D still outperforms CoTracker3 in 2D tracking on ADT dataset with GT depth.
>
> | Methods               | AJ-2D | APD-2D | AJ-3D | APD-3D |
> |-----------------------|-------|--------|-------|--------|
> | Cotracker3 + GT        | 55.3  | 73.1   | 49.6  | 64.0   |
> | DELTA + GT        | 58.9  | 72.4   | 46.9  | 58.8   |
> | TAPIP3D-camera + GT    | **63.1**  | **79.5**   | **61.2**  | **76.5**   |
>
> **2. Evaluation with Sparse Depth on the Waymo Dataset**
>
>
> We created a benchmark consisting of 50 sequences from the Waymo Open Dataset [3] (a subset of TAPVid-3D), paired with depth map annotations obtained by densifying sparse LiDAR depth using a simple nearest-neighbor approach, as proposed in the DriveTrack [4] paper. We report the performance of TAPIP3D-camera, DELTA, and CoTracker3 on this dataset.
>
> | Methods               | AJ-2D | APD-2D | AJ-3D | APD-3D |
> |-----------------------|-------|--------|-------|--------|
> | Cotracker3 + LiDAR       |  63.7     |   78.8     |   16.5     |   24.1    |
> | DELTA + LiDAR       |  **71.4**     |     **85.0**   |   21.0      | 30.8     |
> | TAPIP3D-camera + LiDAR   |  67.7     |     82.2  |   **21.8**    |   **31.6**     |
>
>
> **3. Explanation for Results on LSFOdyssey**
>
>  (1) ***Substantial performance gap between TAPIP-3D and its competitors in LSFOdyssey***
> The LSFOdyssey dataset features challenging scenarios with many objects rotating and occluding each other, where knowing the accurate local 3d geometry help substantially.
>
>
> (2) ***Why does SpatialTracker's performance in LSFOdyssey drop when using ground truth depth compared to MegaSAM predictions?***
> In the provided GT depth, the background has large depth values, and SpatialTracker normalizes all valid depths to the range [0,1], which results in compressing the scene to a small region in SpatialTracker’s tri-planes. Our model is more robust to large depth ranges because it does not require depth values to fall within a fixed interval.
>
> We further fixed the depth value issues of the background regions in SpatialTracker's code by setting it to zero (previously is a large value) to mark it as an invalid region and re-evaluate SpatialTrack using GT depth. The result on the LSFOdyssey benchmark is below. Although the AJ-3D performance of SpaTracker got improved, it still has a significant gap to our method.
>
>
> | Methods                | AJ-2D | APD-2D | AJ-3D | APD-3D |
> |------------------------|-------|--------|-------|--------|
> | SpaTracker + M-SaM     | 73.8  | 88.5   | 14.8  | 22.8   |
> | SpaTracker + GT        | 71.2  | 87.7   | 24.1  | 34.5   |
> | TAPIP3D-world + GT     | **78.5**  | **92.8**   | **72.2**  | **85.8**   |
>
> **4. Clarification on Evaluation Metrics (definition of APD)**
>
> ***Use depth-adaptive APD as a tracking metric, as introduced in TAP-Vid-3D***
>
> This is a misunderstanding. APD_3D stands for a depth-adaptive variant that scales the threshold based on object distance, precisely as introduced in TAPVid-3D [2]. So, we do use the metric you are suggesting. APD_2D is a fixed-threshold metric computed in pixel space, following [1].
>
> We apologize for the confusion regarding APD metrics. We will include the formal definitions in the text to make this distinction clear.
>
> ***The notation 'T' is used to denote the number of frames in a video and  to represent the size of a local window.***
>
> We agree the notation could be clearer. In the final version, we will modify $T$ (temporal window length) to $W$ to avoid confusion with the total number of video frames.
>
>
> **5. DELTA Failure Cases (Figure 4)**
>
> DELTA is trained with depth augmentation. As a result, it sometimes “corrects” geometry even when it conflicts slightly with the input depth. By contrast, our model always respects the given geometry.
>
>
> [1] TAP-Vid: A Benchmark for Tracking Any Point in a Video. NeurIPS, 2022.
>
>
> [2] TAPVid-3D: A Benchmark for Tracking Any Point in 3D. NeurIPS, 2024.
>
> [3] Scalability in Perception for Autonomous Driving: Waymo Open Dataset. CVPR, 2020.
>
>
> [4] DriveTrack: A Benchmark for Long-Range Point Tracking in Real-World Videos. CVPR, 2024.

---

> > ### Comment · Reviewer_Sbrq · 2025-08-05
> >
> > Thank you for your comprehensive rebuttal. All of my initial concerns have been resolved with the impressive results the authors have presented in the TAPVid-3D ADT. While the other reviewers have some concerns about the vulnerability to camera or depth noise, I believe the field of camera pose and depth estimation is growing surprisingly fast, supported by recent works such as VGGT [1], CUT3R [2], and pi-3 [3]. I'm confident that these concerns will diminish in the near future, making this work even more important as TAPIP-3D stands to benefit greatly from those advancements.
> >
> > I am now a stronger supporter of this paper and have raised my scores accordingly.
> >
> > [1] Wang et al., "VGGT: Visual Geometry Grounded Transformer", CVPR'25
> >
> > [2] Wang et al. "Continuous 3D Perception Model with Persistent State", CVPR'25
> >
> > [3] $\pi^3$: Scalable Permutation-Equivariant Visual Geometry Learning,

---

> > > ### Author Response · Authors · 2025-08-08
> > > **Response to Reviewer Sbrq**
> > >
> > > We sincerely thank Reviewer Sbrq for the thoughtful comments and for recognizing that the initial concerns have been fully addressed. We’re also grateful for the encouraging remarks about the significance of our work and its potential to benefit from ongoing advances in camera pose and depth estimation.

---

### Official Review · Reviewer_4cTm · 2025-07-03

**Clarity:** 4
**Significance:** 3
**Originality:** 3
**Rating:** 5
**Confidence:** 3

**Summary:**

The paper transforms the 3D tracking problem from a frame-to-frame 2D problem into a frame-to-model 3D tracking approach. This method effectively decouples object motion and camera motion in the scene to simplify the problem. It employs a novel neighborhood attention mechanism instead of traditional matching methods to optimize the entire 3D-based tracking approach. Through extensive qualitative and quantitative analysis on datasets, each designed component significantly enhances the overall system while maintaining good real-time performance.

**Questions:**

Have you investigated the impact of varying degrees of input image blur on the system? For example, motion blur caused by camera or object movement, or severe blur due to defocus.

Most of the qualitative results focus on objects with significant motion in the foreground. Have you explored whether robust 3D tracking can be achieved for objects at a far distance with significant blur, such as people in the audience seats in a football field?

**Ethical Concerns:**

["NO or VERY MINOR ethics concerns only"]

**Final Justification:**

Overall, this is a good paper with solid contributions and good experimental validation which merit publication. The authors addressed the reviewers concerns well in the rebuttal. Therefore, I would raise the score to 5.
I trust the authors to incorporate as much reviewer feedback as possible into the final version.

**Limitations:**

No, I believe that, beyond the illumination flickering and certain depth map dependency discussed by the authors, this work also has the limitation of struggling to achieve accurate trackig under severe motion blur. Additionally, depth maps estimated from monocular inputs are affected by blur, leading to erroneous edge estimation.

**Paper Formatting Concerns:**

No major formatting issues.

**Quality:**

3

**Strengths And Weaknesses:**

Paper Strengths:

1. The paper achieves more efficient optimization by decoupling the representation of tracking processing from camera motion, while simplifying the complexity of the target function in high-dimensional spaces.

2. The paper enhances tracking consistency in 3D space through cross-attention in 3D neighborhoods.

3. The proposed system exhibits certain robustness to occlusions and can benefit from the quality of depth information.

Major Weaknesses:

1. The paper mentions that systems like NeRF incur significant computational overhead but does not provide a detailed comparison. To my knowledge, 4DGS [1] can achieve real-time performance at 82 FPS with good results.

2. Regarding robustness, the paper does not explicitly investigate the impact of blurry input images on the overall system. It only explores the system’s usability under different depth conditions and low-light scenarios.

[1] G. Wu, T. Yi, J. Fang, L. Xie, X. Zhang, W. Wei, W. Liu, Q. Tian, and X. Wang, “4D Gaussian splatting for real-time dynamic scene rendering,” in Proc. IEEE/CVF Conf. Comput. Vis. Pattern Recognit. (CVPR), Jun. 2024, pp. 20310–20320.

---

> ### Author Rebuttal · Authors · 2025-07-31
>
> ### Response to Reviewer 4cTm
>
> We thank the reviewer for the thoughtful and detailed feedback, and for highlighting the strengths of our decoupling strategy and 3D N2N attention mechanism.
>
> **1. Efficiency Comparison with Optimization-Based Methods**
>
> We acknowledge the concern regarding efficiency comparisons with optimization-based scene representation methods such as 4DGS [1] and appreciate the opportunity to clarify.
>
> While 4DGS reports high rendering frame rates of 82 FPS, this only reflects only its rendering speed after training. The method requires time-consuming per-sequence optimization, with training times reaching up to 30 minutes per sequence on the HyperNeRF dataset, as reported in Table 2 of [1]. Similar limitations apply to other optimization-based methods such as NeRF.
>
> In contrast, TAPIP-3D performs long-term point tracking in a fully feedforward manner, running at approximately 10 FPS without any per-sequence training (fitting).
>
> Given the significant differences in computational cost and target applications, direct efficiency comparisons between these methods are not informative in practice. We will clarify this important distinction in the next version of the paper.
>
>
> **2. Robustness to Blur**
>
>
> **2.1 Impact of Blur at Varying Degrees**
>
> Our method augments the training data by introducing motion blur and randomly applying Gaussian blur to both images and depth maps to improve robustness against blurry inputs. We also adopt all the 2D data augmentations from CoTracker3 as mentioned in L10 of the Supp. file, which includes gaussian blur to images. We will detail it in the paper.
> As a result, our model can tolerate some degree of blur.
> To evaluate the model’s robustness more concretely, we present a table below showing quantitative results of TAPIP3D-world + M-SaM on the LSFOdyssey dataset with varying levels of gaussian blur applied:
>
> | σ (Blur Level) | Method   | APD-3D     | APD-2D     |
> |----------------|----------|------------|------------|
> | 0 (No Blur)    | TAPIP-3D (ours) | 29.2       | 87.5       |
> |                | DELTA    | 28.0       | 90.2       |
> | 0.5            | TAPIP-3D (ours) | 21.2       | 87.4       |
> |                | DELTA    | 20.3       | 90.3       |
> | 1.0            | TAPIP-3D (ours) | 20.6       | 86.6       |
> |                | DELTA    | 19.8       | 90.3       |
> | 2.0            | TAPIP-3D (ours) | 18.1       | 86.0       |
> |                | DELTA    | 17.5          | 90.2          |
>
> Note that in the table above, our model's 3D performance surpasses the baseline, DELTA, at every blur level, demonstrating competitive robustness. From the visualizations, we observe that with blurry inputs, the points align reasonably well with objets, while depth estimates of the same object become less consistent across frames, possibly due to blurry inputs being out-of-distribution for MegaSAM.
>
> **2.2 Tracking Blurry and Distant Objects**
>
> We acknowledge that low-resolution and distant objects may lead to degraded geometry, particularly around edges. To mitigate artifacts from edge interpolation caused by depth estimation, we apply a simple edge-removal preprocessing step before feeding the depth maps into our pipeline. This approach is commonly adopted in 3D visualization for depth models.
>
> With this preprocessing in place, we observe that on DriveTrack (a subset of TAPVid-3D), where vehicles often appear small, blurry, and distant, our model is often able to track points on these objects despite the sparse pixel density after unprojection.
> However, we acknowledge some failure cases, particularly when two distant vehicles occlude each other. In such scenarios, the estimated depth geometry may become ambiguous, with disconnected objects appearing fused, making it hard for our model to distinguish between them. We believe this issue is related to the low resolution of the depth model (MegaSAM operates at 192×256 and is upscaled prior to use in our model). We expect that future improvements in foundational geometry (depth and camera pose)  estimation models will help mitigate these challenges. We will include a discussion of these limitations in the final version of the paper.
>
>
>
> ----------
>
> **References:**
>
> [1]  Wu, T. Yi, J. Fang, L. Xie, X. Zhang, W. Wei, W. Liu, Q. Tian, and X. Wang, “4D Gaussian splatting for real-time dynamic scene rendering,” in Proc. IEEE/CVF Conf. Comput. Vis. Pattern Recognit. (CVPR), Jun. 2024, pp. 20310–20320.

---

> > ### Comment · Reviewer_4cTm · 2025-08-07
> > **Rebuttal response**
> >
> > I thank the authors for providing additional clarifications and experiments. I have also reviewed all other comments.
> > Regarding real-time performance, the authors have meticulously clarified that GS and NeRF-based methods require substantial computational resources and per-scene fitting during training, demonstrating the competitive real-time performance of this work.
> > In other aspects, the authors have effectively analyzed limitations concerning blurry inputs and low-resolution objects, conducting thorough comparisons in the blurry input section to show that their proposed pipeline better handles such inputs. However, two questions remain:
> > - It is unclear whether the improvement is due to MegaSAM or the model trained with augmented data.
> > In the test experiments, the authors only applied Gaussian blur to the dataset to evaluate the method, without testing the impact of motion blur through multi-frame averaging.
> > - The method performs well in the absence of additional sensor noise and could benefit from advancements in foundational geometry estimation. However, in cases of severe sensor blur, the method’s lower robustness to blurry noise or noise from MegaSAM may limit its performance. Designing a refinement module for MegaSAM to address this limitation could potentially enhance performance.
> > Overall, this is a highly substantial and outstanding work, pioneering the use of tracking in 3D space to simplify the problem. I am considering raising my score.

---

> ### Author Response · Authors · 2025-08-08
> **Response to Reviewer 4cTm**
>
> Thank you for the follow-up. We are very pleased to hear that you are considering raising your score.
> 1. Since we give all methods the same pose/depth estimators, this cannot be the source of relative improvement.
> 2. We are not sure what you mean by the detail about “augmented data”. The baselines (DELTA, SpatialTracker) use largely similar training time augmentations as TAPIP3D.
> 4. We can also add an experiment with multi-frame averaging, for a more accurate assessment of the robustness to motion blur. We agree this would be more accurate than Gaussian blur.
> 4. Potential refinements to MegaSAM: We agree! This would be exciting to explore in future work.

---

### Note · Authors · 2025-08-13

We sincerely thank all reviewers for their thoughtful feedback. Below we (1) summarize strengths highlighted by reviewers, (2) address the key concerns, (3) commit to concrete revisions, and (4) summarize with our broader contributions.

### 1) Reviewers’ highlighted strengths
- **World-frame tracking.** We remove camera motion via extrinsics and track directly in a global XYZ frame rather than camera-centric UVD, simplifying trajectories and improving accuracy under large camera motion.
- **3D Neighborhood-to-Neighborhood Attention.** Our **3D N2N Attention** replaces grid correlation volumes and operates on unstructured 3D neighborhoods, exploiting true geometric context and naturally supporting stabilized world-frame tracking.
- **Novelty and motivation.** Reviewers noted the clear motivation and originality of leveraging geometry for long-term 3D tracking; Reviewer Sbrq emphasized our shift away from over-reliance on 2D cues.

### 2) Main concerns and our responses
- **Robustness to geometry quality (depth & poses).** We evaluated multiple depth estimators (MegaSAM, UniDepth), paired **GT depth** with distinct pose estimators (CUT3R, MegaSAM), and injected angular pose noise. **TAPIP3D** maintains strong performance across these settings.
- **Sparse/real-world depth.** Using LiDAR on the Waymo dataset, we observe competitive 2D and 3D results, showing practical applicability with sparse depth.
- **Blur robustness.** Gaussian-blur experiments (motion/distance) show TAPIP3D degrades more slowly than baselines, indicating resilience to common image degradations.

### 3) Revisions we will include in the final paper version:
- Expand 2D baselines and **TAP-Vid** comparisons.
- Ablate depth/pose quality, including sparse LiDAR depth.
- Clarify metric definitions and notation.
- Detail runtime/efficiency and distinguish from optimization-based trackers.
- Add further blur-robustness analyses.

### 4) Contribution
We appreciate the reviewers’ recognition of TAPIP3D as a promising direction. Our method achieves strong 3D point tracking with current monocular or sensor-depth video inputs and **improve as depth/pose estimation advances**. We believe our contributions—a **world-frame formulation** for long-term 3D point tracking and a **geometry-aware attention mechanism** for correspondence—offer a useful building block for broader 3D perception in robotics and autonomous driving, and will stimulate robust, geometry-driven tracking research.

---

### Decision · Program_Chairs · 2025-09-17

**Decision:**

Accept (poster)

**Comment:**

After the discussion phase, three reviewers recommend the paper, including one who does so very strongly.  The fourth	 reviewer still has reservations about the experimental validation but I believe the authors did a good job of addressing his concerns in their rebuttal. I therefore recommend acceptance.